# Entropy Analysis of Protein Sequences Reveals a Hierarchical Organization

**DOI:** 10.3390/e23121647

**Published:** 2021-12-07

**Authors:** Anastasia A. Anashkina, Irina Yu. Petrushanko, Rustam H. Ziganshin, Yuriy L. Orlov, Alexei N. Nekrasov

**Affiliations:** 1Engelhardt Institute of Molecular Biology, Russian Academy of Sciences, Vavilov St. 32, 119991 Moscow, Russia; irina-pva@mail.ru; 2Shemyakin-Ovchinnikov Institute of Bioorganic Chemistry, The Russian Academy of Sciences, Miklukho-Maklaya St. 16/10, 117997 Moscow, Russia; ziganshin@mail.ru (R.H.Z.); alexei_nekrasov@mail.ru (A.N.N.); 3The Digital Health Institute, I.M. Sechenov First Moscow State Medical University of the Ministry of Health of the Russian Federation (Sechenov University), Trubetskaya 8-2, 119991 Moscow, Russia; orlov@d-health.institute; 4Agrarian and Technological Institute, Peoples’ Friendship University of Russia (RUDN University), Miklukho-Maklaya Str. 6, 117198 Moscow, Russia

**Keywords:** protein structure, hierarchy, protein sequences, ANIS method, informational structure, protein design, foldon, peroxiredoxin, interleukin 13, hydrolases, oligopeptidase B, TNF, HSP70, carboxypeptidase, hem-containing proteins

## Abstract

Background: Analyzing the local sequence content in proteins, earlier we found that amino acid residue frequencies differ on various distances between amino acid positions in the sequence, assuming the existence of structural units. Methods: We used informational entropy of protein sequences to find that the structural unit of proteins is a block of adjacent amino acid residues—“information unit”. The ANIS (ANalysis of Informational Structure) method uses these information units for revealing hierarchically organized Elements of the Information Structure (ELIS) in amino acid sequences. Results: The developed mathematical apparatus gives stable results on the structural unit description even with a significant variation in the parameters. The optimal length of the information unit is five, and the number of allowed substitutions is one. Examples of the application of the method for the design of protein molecules, intermolecular interactions analysis, and the study of the mechanisms of functioning of protein molecular machines are given. Conclusions: ANIS method makes it possible not only to analyze native proteins but also to design artificial polypeptide chains with a given spatial organization and, possibly, function.

## 1. Introduction

In 1961, Anfinsen wrote [1] that the amino acid sequence defines proteins’ structural and functional properties. Multiple attempts to identify the laws in the arrangement of amino acid residues in protein sequence did not give significant results [2,3,4]. In particular, Szoniec and Ogorzalek showed that only 1% of residues in a protein sequence are non-random [4]. So, there is a contradiction: physicochemical characteristics of amino acids along the polypeptide chain determine properties of the protein, but amino acid residues are arranged in sequence almost randomly. This contradiction indicated that the description of protein sequences as a polymer consisting of amino acid residues is inadequate. Having conceived to create an adequate explanation, we tried to find such an approximation that made it possible to obtain the local characteristics of the polypeptide chain from combinations of statistical parameters.

Analyzing the context (environment) of different types of amino acids in protein sequences, we found that the frequency of amino acid residue occurrence differs at various distances between amino acids in the sequence. In addition, the occurrence frequencies of amino acids to the right and the left of the given position of amino acid residue are not symmetric [5]. After analyzing the frequency of occurrence of amino acid residues in the immediate environment (±20 residues) and comparing them with each other, it turned out that the context of some amino acids looks more similar. Thus, we found three groups of similar amino acid residues. There are amino acids V, I, L, F, Y in the first group, K, R, Q, E in the second, and S, T, D, N, A in the third group. Finally, there were a set of amino acid residues (M, C, P, G, H, W), the surroundings of which were less similar to other residues [5].

Carefully studying the sequence context of amino acids, we concluded that residues with similar physicochemical properties had a similar environment. These observations made it possible to substantiate the relationship between the local occurrence of amino acid residues around an amino acid and its physicochemical properties. We hypothesized that such a correlation of the amino acid residue with its local environment is a distinct feature of natural protein sequences that have undergone molecular evolution. This work led to the idea that it is necessary to consider the fragments of the protein sequence as “blocks” in the organization of the protein sequences to describe the physicochemical properties of the polypeptide chain [5]. We studied the entropy characteristics of non-homologous sets of protein sequences to determine the optimal size of such blocks [6]. We calculated the matrices of amino acid residues frequency of occurrence at varying distances in a sequence for three sets of sequences of natural proteins. Shannon’s entropy for such matrices, depending on the distance between the residues, has an S-shape with a pronounced oscillatory component. The minimum values of Shannon’s entropy correspond to the distance of three amino acids, which corresponds to fragments of the polypeptide chain by length 5. This fact made it possible to assert that peptide blocks of five amino acid residues are optimal for studying the structural organization of protein sequences.

We proposed to consider a block of five amino acid residues as a single unit. We called it “information unit”, and the method of sequence analysis itself was called ANIS (ANalysis of Informational Structure) [6]. If we attribute its occurrence in a sufficiently large database of natural protein sequences to a pentapeptide, we can see the most conserved fragments. By allowing possible substitutions in the pentapeptide sequence when counting for its occurrence, we find variants with all possible mutations and increase its evolutionary depth. A special mathematical method could reveal a hierarchy in the pentapeptide frequency curve [7]. The entire ANIS method is described in detail below in the Materials and Methods section.

This work continues the cycle of our works devoted to the analysis of protein sequences [6,7] based on the information estimates. Here we will consider the effect of the information unit size and the number of permissible substitutions on the quality of the information graph of the sequence. In this work, we show that the developed mathematical apparatus gives stable results even for a significant variation in the sizes of fragments and the accuracy of their correspondence.

## 2. Materials and Methods

Figure 1 shows the scheme of protein sequence analysis implemented in work [6]. The protein sequence dataset for this work should consist of non-homologous protein sequences and be large enough. Generating such a dataset is itself a challenging scientific task. For this reason, the second parts of the various releases of the PIR database have been used. This part included qualitatively defined sequences of proteins that did not show a significant level of similarity with previously obtained and characterized protein sequences. As a result, the second parts of the releases 18, 27, and 49 of the PIR database were used as protein sequence datasets, which included 5556 sequences (1,510,026 amino acid residues), 12,607 sequences (3,417,043 amino acid residues), and 58,089 sequences (21,699,210 amino acid residues), respectively.

Suppose we have a protein sequence of ***L*** amino acid residues length. Let us calculate the matrices ***F^k^*** (20 × 20) of amino acid pairs occurrences, which contain the frequencies of occurrence of residues separated in the sequence by ***k*** positions (***k*** = 0,…,40). The position of the first amino acid residue in the pair shifts in series from the first amino acid residue of the sequence to ***L-k*** residue during calculation. The ***F^k^*** matrices summarize data from all sequences from the protein sequence database. Thus, the matrices ***F^k^*** are characteristics of the entire set of the sequence database. For further analysis, we transform the matrices ***F^k^*** of occurrence of frequencies into matrices of the probability of occurrence ***P^k^*** of pairs of amino acid residues, separated in sequences by ***k*** residues. 

For each of the probability matrices ***P^k^***, we can calculate the amount of informational entropy ***S^k^*** contained in it by the Formula (1) [8]:(1)$$S^{k}=-\sum_{i=1}^{20}\sum_{j-1}^{20}P_{i,j}^{k}\log_{{}_{2}}P_{i,j}^{k}$$

The values of information entropy ***S^k^*** can be represented as a graph of the dependence on the distance ***k*** between amino acid residues. The absolute values of the changes in information entropy over the interval ***k*** from 0 to 40 are insignificant. To reveal informational entropy dependence on the distance ***k*** between amino acid residues and to reduce the influence of the size of the protein sequence datasets, we normalized information entropy ***S^k^*** by the value of ***S*****^0^** since the adjacent amino acid residues in the sequence (at ***k***
*=* 0) are maximally correlated with each other.

One can see (Figure 2A) that the informational entropy dependences for three different sets of protein sequences have an identical form and a pronounced oscillatory component, i.e., they are stable integral characteristics of sets of protein sequences.

Fourier analysis of the dependences in the R package revealed two periods in the oscillatory component—2.9 and 3.6. These values correspond to two classical elements of the secondary structure—helix 3_10_ and α-helix, which were first described by L. Pauling [9]. It is interesting to note that no β-structure related periodicity was found.

We removed the oscillatory component from the obtained dependencies (Figure 2A) by subtracting the oscillatory curve with the calculated amplitudes in the R package, then the curves took the following form (Figure 2B). Note that the S-shape is common to all three datasets. It can be seen that the value of the normalized informational entropy ***S^k^/S*^0^** depends on the size of the protein sequence dataset. The higher the number of protein sequences in the dataset, the higher the value of normalized information entropy ***S^k^/S*^0^**. Figure 2B shows that as the distance between amino acids increases, the correlation decreases. The tendency for the ***S^k^/S*^0^** value to stabilize at a distance greater than 40 means that there is a small correlation at large distances.

The lowest level of normalized informational entropy ***S^k^/S*^0^** is observed at ***k*** = 3, which corresponds to fragments of the polypeptide chain of five amino acid residues long. This suggests that pentapeptides are optimal for studying the structural organization of protein sequences. We proposed to consider blocks of five amino acid residues as a whole unit and called a peptide block of this length an “information unit”.

The use of this basis unit of protein sequences made it possible to propose a new method of analysis to reveal the hierarchical organization in the protein sequences. The method consists of several steps:1.The protein sequence is dissected on overlapping blocks of five adjacent amino acid residues, which are obtained by shifting one by one position of a frame of five residues from the N to the C-end of the sequence;2.The frequencies of occurrence ***f_i_*** of each block in the sequences of a non-homologous database (large enough) are calculated. This stage is shown in Figure 3A;3.The protein population profile ***F*** of the sequence is constructed as follows. Each amino acid residue is assigned with the sum of the frequencies of occurrence of all five-member blocks in which it is included. This step is shown in Figure 3B, where the resulting population profile of the studied sequence is represented by a black line. For the residue at position ***i***, the value in the population profile will be
(2)$$F_{i}=\sum_{n=i-2}^{n=i+2}f_{n}$$
4.At this step, the population profile ***F*** obtained in the previous step is analyzed by inscribing centrosymmetric smoothing functions ***φ(i,a)*** whose centers correspond to all possible positions, having different widths ***a*** at half height (Figure 4). For a protein of ***L*** amino acid residues, the centers ***i*** can be amino acid residues with numbers from ***b/***2 + 1 to ***L***
*− **b/***2 if the smoothing functions have a base ***b***.

Figure 5A shows the dependence of the area under the inscribed smoothing function on the amino acid residue number (from 1 to ***L***) and the width of the smoothing function ***a*** at half height. The color of each point in Figure 5 corresponds to the area under the smoothing function. The darker areas correspond to higher values under the smoothing functions. The scope of the function is triangular, since the larger the value of ***b***, the fewer points can be obtained. The surface in Figure 5A can be represented in the form of a tree-like graph (Figure 5B), which is more convenient for further practical use.

The proposed method for the analysis of sequences was called the method of ANalysis of Information Structure (method ANIS). The set of hierarchical graphs represent the informational structure (IS) of a protein. A separate hierarchical graph or a given fragment of a graph is an ELement of the Information Structure (ELIS) of a protein. ELIS is characterized by its position in the amino acid sequence and rank—the number of element fusions observed within the ELIS plus one. The lowest level of ELIS (without branch points) has a rank of one.

## 3. Results

Although we believe that the size of the “information unit” of five amino acid residues is optimal, similar results can be obtained with other sizes **λ** of the “information unit” and the number **δ** of allowed substitutions when calculating the occurrence of the peptide in the protein sequence database (Figure 6).

Figure 6 shows that the IS in the protein sequence can be revealed in a fairly wide range of the length of the information unit and the number of allowed substitutions. So, we considered the length of the information unit **λ** from 2 to 8 amino acid residues, and the number of permissible substitutions **δ** from 0 to 6. With the size of the information unit **λ** = 8 amino acid residues in the Q9GZZ6 sequence, not all octapeptides were found in the NRDB90 protein sequence database at least once. As a result, when calculating the IS, a gap arises, which leads to a decrease in the area of definition of the smoothing function ***φ (i, a)*** (Figure 6, **λ** = 8, **δ** = 0). When calculating the occurrence of octapeptides with allowed substitutions (Figure 6, **λ** = 8, **δ** > 0), the area of definition does not decrease. Thus, with the size of the information unit **λ** = 8, the number of mismatches should be **δ** > 0.

The figure shows that, depending on the value of **λ** and **δ**, the IS of the protein sequence changes. The changes concern both the location of the first rank ELISes and the hierarchical graph as a whole. Moreover, changes in the location and number of ELIS of the first rank are not as noticeable as changes in the hierarchical graph as a whole. With an increase in the size of the information unit **λ**, we see a sequential addition of separate branches into hierarchical graphs. It occurs due to taking into account the correlation between distant amino acid residues in the chain. This is clearly seen if we consider the right vertical column from top to bottom in Figure 6, with a constant value of **δ** = 0 and an increase in **λ** from 2 to 8. A small increase in **δ** leads to an improvement in the clarity of the hierarchical graph picture. With a further increase in **δ**, the hierarchical graph splits into separate branches from ELIS of the first rank. The most optimal, from our point of view, is the length of the information unit **λ** = 5, and the number of allowed substitutions **δ** = 1. We will use these parameters further to study the hierarchy of protein sequences.

Considering the arrangement of the lowest-level elements (first-rank ELISes) in different protein sequences, we noticed that they are located at different distances from each other. The protein sequence can be divided into three types of sites (Figure 7). The first type of sites, NORMAL sites—when the distance between the first rank ELISes is commensurate with the size of the information unit (for pentapeptides, the distance between the centers of first rank ELISes is 5 ± 1 amino acid residues) (Figure 7A). The second type is when the first rank ELISes overlap in the sequence. The distance between the centers of first rank ELISes for pentapeptides is from 1 to 4 amino acid residues. We called this type of site with a high density of distribution Abnormal Distribution Density+ sites (ADD+) (Figure 7B). The third type is when the distance between the centers of first rank ELIS is significantly greater than the size of the information unit (ADD− sites) (Figure 7C) [10].

This differentiation of the protein sequence into sites with different densities of the first rank ELIS makes it possible to characterize different parts of the polypeptide chain by the degree of flexibility of its backbone. We assumed that ADD+ sites are the least flexible, while ADD− sites have a flexible polypeptide backbone. Based on these assumptions, we were able to explain a number of properties of proteins and protein complexes. For example, in the interfaces of enzyme-inhibitor complexes, one of the proteins contains at least 75% of amino acid residues in ADD− sites [10]. The flexibility of one of the interacting protein chains ensures efficient interaction between amino acid residues at the interface. The choice of sites containing positively charged amino acid residues Arg and Lys of ADD+ sites as B-epitopes increases their immunogenicity several times, which can be used in the development of vaccines [11,12]. An analysis of the location of ADD+ and ADD− sites in the spatial structure of proteins made it possible to propose a model for the functioning of hem-containing proteins [13]. In the heat shock protein sequence, ANIS method isolated a peptide that activates the production of normal killers (NK cells) [7]. These examples are discussed in more detail below.

In order to understand whether such a short protein fragment of five residues long has a predominant conformational state, we performed a large-scale molecular dynamics study of model pentapeptides [14]. We in series replaced amino acid residues in the same structure of poly-A pentapeptide by a special scheme. Thus, we have generated 44,860 pentapeptide structures in the same conformation with different sequences. Molecular dynamics simulations were carried out in a potential field with the implicit solvent for 10 ps for each pentapeptide structure. Analysis of the last 5000 conformations of each molecular dynamics trajectory showed that 1225 of 44,860 (2.73%) structures had a stable conformation in which the peptide was at least 80% of the modeling time (Figure 8). A similar calculation for the human Na, K ATPase protein (Uniprot code P05024) with 1017 overlapping pentapeptides revealed 79 pentapeptides (7.67%) with stable conformations, which overlapped 32.4% of the Na, K ATPase protein sequence. 

The 3D structures of stable peptides found by molecular dynamics have a number of similar conformations. The MaxCluster program (http://www.sbg.bio.ic.ac.uk/maxcluster/, accessed on 1 October 2018) was used to combine them into 54 groups according to the criterion of spatial structure similarity. Figure 9 shows the central conformations of the 20 largest clusters of stable pentapeptides.

In the stable pentapeptide, fragments corresponding to its beginning, middle and end can be distinguished, each of which consists of three amino acid residues. These fragments can have conformations corresponding to the elements of the secondary structure (α−helices (**α**) or extended (**e**)). Then each pentapeptide can be described by three letters, for example, **ααα** or **e****αα**. Let us consider the functional role of each type. Peptides **αee** or **ααe** are peptides that terminate the alpha helix. The peptide **ααα** is a peptide that continues (maintains) the alpha helix. Peptides **e****αα** or **ee****α** start an alpha helix. Peptide **αe****α** forms a turn or bend of the alpha-helix. Similar designations can be done for unfolded structures. In our work [14], all 54 found classes of structurally stable peptides were classified according to their possible role in the formation of secondary structure elements. 

The ANIS method, due to its algorithm, reveals hierarchical elements formed in the process of molecular evolution. Apparently, in such a hierarchy, the factors by which the selection took place should be realized. Let us note two facts revealed during the application of the ANIS method to various proteins. First, in some proteins, ELISes of the highest rank coincide with structural domains (for example: pepsin, cytidine deaminase, lymphocyte adhesion glycoprotein, bacteriophage t4 viral protein, etc.). Second, in a number of enzymes, the catalytic center residues are distributed over different ELISes of the highest rank.

These facts made it possible to make three assumptions:Elements of the spatial structure corresponding to ELISes can independently form their spatial structure;Elements of the spatial structure corresponding to ELISes exhibit characteristics close to those of structural domains;The efficiency of the interaction of sites within a protein globule and interactions of interfaces in protein–protein complexes depends on the density of the first rank ELISes in the interacting sites.

These assumptions allowed us to propose a number of practical applications for the ANIS method. First, a protein design method in which fragments corresponding to ELISes can be removed from native protein sequences. It is assumed that in this case the native mechanism of folding of the protein molecule undergoes minimal disturbances. Second, we proposed a method for editing sites of intermolecular interaction. We can locally change the ability of the polypeptide chain for adaptive conformational rearrangements by changing the amino acid sequence. These methods have been tested in a number of experimental studies, which can be roughly divided into six directions: protein design, intermolecular interactions, identification of immunogenic sites in proteins, studying the mechanisms of protein functions, the functional role of protein structures corresponding to the highest rank ELISes, and identification of minimal functional protein fragments.

### 3.1. Protein Design

#### 3.1.1. Human 1-CYS Peroxiredoxin (*h*PrxVI)

The ANIS method was applied to obtain a truncated active form of Human 1-CYS peroxiredoxin for medical use [15]. This protein was proposed as a constituent part of pulmonary therapy drugs. In order to avoid possible side effects, particularly immune reaction to exogenous protein, and to simplify the isolation and purification, full-sized enzymes can be advantageously replaced by their fragments retaining some of the selected biochemical activity and specifically antioxidant properties. It was known from experimental data that the active center is located in the N-terminal part of the molecule. After analyzing the protein sequence by the ANIS method, it was proposed to remove one or two regions containing detached high-rank ELISes from the C-terminus (Figure 10). Removal of one of these regions made it possible to obtain a protein shortened by 21%, retaining 95% of the native enzymatic activity [15]. We interpret the retention of enzymatic activity as the stability of the protein to fragment removal.

#### 3.1.2. Human Interleukin 13 (*h*IL13)

Interleukin-13 (*h*IL13) is one of the cytokines involved in the development of Th2-type immune response. It plays an important role in the pathogenesis of asthma and other allergic diseases. The aim of work [16] was to obtain an antagonist of natural interleukin-13, which blocks the development of an allergic reaction. The calculation of the IS revealed the four highest rank ELISes in the *h*IL13 sequence (Figure 11). One of them has only one first rank ELIS and consists of 9 amino acid residues in the central part of the sequence (Figure 11D, in orange). Deletion of this peptide from the protein resulted in recombinant protein, which ceased to cause an allergic reaction while maintaining all its other functions [16].

Figure 12 depicts the elements of the spatial structure corresponding to the ELISes from Figure 11. Three of the four elements of the spatial structure are not classical elements of a secondary or super-secondary structure. Only the structure corresponding to ELIS 3 is a hairpin of two alpha-helices. ELIS 1 and 4 also contain alpha-helices, but they are complemented by adjoining loop-like structures. ELIS 2 includes a single ELIS of the first rank in the IS, and in the spatial structure, it corresponds to a turn of the polypeptide chain (Figure 12C). We assumed that each turn of the polypeptide chain corresponds to each ELIS of the first rank. However, there are many more turns of the polypeptide chain in the spatial structures corresponding to ELIS 1, 3, and 4. We suggested that, in addition to the turns of the polypeptide chain, which are caused by the local amino acid sequence, the spatial structure can contain turns of the polypeptide chain, which were formed during the collapse of the polypeptide chain from the pre-folding conformational state to the folded state. So, we believe that all first rank ELISes are turns of different types, but not all turns are first rank ELISes. 

#### 3.1.3. Oligopeptidase B from *Serratia proteamaculans*

In order to change the substrate specificity of oligopeptidase B from *Serratia proteamaculans* (PSP) in the paper [17] we analyzed the IS of its sequence. The natural form of the enzyme hydrolyzes only short substrates. Calculation of the IS of the enzyme made reveals six ELISes of the highest rank in its sequence. The N-terminal region, which corresponds to the highest rank ELIS (Figure 13) from the oligopeptidase spatial structure envelops the catalytic domain of PSP and supposedly prevents hydrolysis of high molecular weight substrates. Elimination of this fragment makes a truncated recombinant form of the enzyme, which can hydrolyze extended peptide chains. The experimentally obtained truncated form of the enzyme showed the desired specificity [17].

### 3.2. Intermolecular Interactions

Based on the local density of first rank ELIS in the protein sequence there are three types of sites (ADD−, ADD+, NORMAL) [10]. So, there are six possible combinations of the sites in protein–protein interaction interfaces, depending on the interaction interface sites types (Table 1).

We studied six complexes of hydrolytic enzymes with their inhibitors of different sources and molecular weight [10], for example, trypsin with bovine pancreatic trypsin inhibitor (Trps/BPTI) and subtilisin with inhibitor (Subt/CI-2A) (Figure 14). The interaction interfaces of hydrolases with inhibitors were analyzed for the density of distribution of the first rank ELIS in the sequence. It turned out that 84.2% of the residues in the interface belong to the composition of ADD sites (from 62% to 96%, depending on the protein) (Table 2).

Based on the data obtained above, we suggested that ADD− regions are required for efficient interaction at the binding site. Such regions are structurally flexible and capable of adaptive conformational rearrangements. We assume that interaction occurs most efficiently in the ADD+/ADD− interfaces. Such interaction interfaces correspond to the “hand-glove” model. 

### 3.3. Identification of Immunogenic Sites in Proteins

We examined interaction interfaces in a variety of proteins, for example, in vascular endothelial growth factor in complex with a neutralizing antibody (Figure 15).

It turned out that in all the studied antigen–antibody complexes, interfaces are formed in ADD+/ADD− way, usually with a structurally rigid antigen (ADD+ site) and a structurally labile antibody (ADD− site). This observation made it possible to propose a method for searching for immunogenic sites in proteins. The search for antigenic determinants in proteins is an important step in the development of recombinant peptide vaccines. Protein epitopes, recognized by either T or B cells, are the best candidates for immunogenic sites in vaccine design due to their hypoallergenic properties and low manufacturing costs compared to full-length proteins [18,19]. 

B cell epitopes usually have size from five to nine amino acid residues; and such epitopes can be recognized directly by immunoglobulins [20]. There are many reports that peptides containing linear B-cell epitopes can induce a neutralizing immune response [21,22]. The purpose of peptide vaccine production is to induce neutralizing IgG antibodies against the pathogen/antigen [23]. Peptide vaccines are preferred over all others due to their versatility and the possibility of combining both T and B cell epitopes from different proteins to create one polypeptide chain, i.e., this allows the creation of a vaccine containing epitopes from several proteins.

Based on the properties of ADD sites, we suggested that they can have different immunogenic properties, and then the ANIS method can be used to find B epitopes in protein sequences. To determine the reliability of our assumption, we analyzed the location of the known B-cell epitopes of lysozyme proteins (HEL), myoglobin (SWM), and tobacco mosaic virus protein (TMVP) [24,25,26] relative to the ADD sites [11]. The analysis of B-cell epitope location has shown that frequently there are positively charged residues R/K. Ten times more epitopes were found in ADD+-sites containing R/K than in ADD+-sites free of R/K and 2.5 times more than it can be on random [11]. 

To test the hypothesis that peptides located at ADD+ sites containing R/K can serve as epitopes, we constructed six recombinant peptides in the form of virus-like particles. Immunization of mice with all constructs elicited a comparable and significant humoral immune response [11]. Regions of filaggrin molecules, which can act as immunoreactive epitopes reacting with antibodies in the sera of Rheumatoid arthritis patients, were found in a similar study [27]. This method, based on the analysis of the IS of protein sequences, requires additional validation. After that, it may be widely used for the development of vaccines. 

### 3.4. Studying the Mechanisms of Protein Functions. Functional Role of Protein 3D Structures Corresponding to Highest Rank ELISes

#### 3.4.1. Pepsin

Catalytic reactions should be accompanied by some conformational rearrangements of the spatial structure of its active center. The active site is often formed at the contact of two structural domains of a protein to allow motility. In this case, the natural mobility of the structural domains relative to each other provides the necessary conformational lability. 

An example of a protein with such a spatial organization is pepsin, the spatial structure of which is shown in Figure 16. It should be noted that residues Asp32 and Asp215, which form the catalytic center of pepsin, are located on different structural domains of the protein. We suggested that highest rank ELISes can play the role of structural domains, providing limited mobility, which accompanies the catalytic process in enzymes (Figure 16). 

#### 3.4.2. Trypsin

It was shown by the number of hydrolases that the residues that form the catalytic centers of enzymes are located on different ELISes of the highest rank even in cases when the ELISes of the highest ranks are not structural domains [29]. As an example, Figure 17 shows the informational and spatial structure of trypsin. It can be seen that the residues of the catalytic center SER195, ASP102, and HIS57 are distributed over different ELISs, and the highest-rank ELISs themselves do not form spatial structures similar to the structural domains of globular proteins. This indicates that the highest rank ELISes, regardless of their structural organization, play the functional role of structural domains.

#### 3.4.3. Ribonuclease A

It is important to note that the distribution of the catalytic residues of the active site over different ELISes of the highest rank is also observed in proteins that are so small that there is no way to talk about their domain structure. An example of such an enzyme is RNase A, which consists of only 102 amino acid residues (Figure 18).

#### 3.4.4. Carboxypeptidase

Consider the sequence and structure of Carboxypeptidase from Bos taurus (Figure 19) [29]. In the active center of this enzyme there is a bound water molecule and a Zn atom. The Zn atom forms coordination bonds with the residues HIS69, HIS196, and GLU72. Note that residues GLU72 and HIS69 belong to the same highest rank ELIS, marked in green in Figure 19. Residue HIS196, located on another the highest rank ELIS (red), also interacts with the Zn atom. The mobility of these ELISes relative to each other leads to the displacement of the Zn atom and the activation of the main catalytic residue GLU270 through a water molecule bound in the catalytic center. Thus, unbalancing a metal atom from equilibrium underlies the functioning of this enzyme. The mobility of structural fragments corresponding to the highest rank ELIS is the result of exposure of the polypeptide chain to molecules of the surrounding solvent. Consequently, the enzyme molecule works by heat energy from the environment.

#### 3.4.5. Human Hemoglobin

X-ray structural analysis data for the oxy and deoxy forms of hemoglobin showed that the protein function is associated with the displacement of the iron atom from the heme plane [30]. The mechanism of this process can be explained by analyzing the IS of this protein (Figure 20). In the hemoglobin alpha chain, histidine 58 and 87 residues interact with the iron atom. IS shows that these residues belong to different highest rank ELIS. One of them (H87) is located in a single ADD− site. Fragments of the globule corresponding to the highest rank ELISes move relative to each other. Conformational rearrangements during this movement are most likely realized due to ADD− site flexibility. The side chain of the H87 residue, which interacts with iron, dislocates the iron atom from the heme plane during the process of this interaction, ensuring the hemoglobin function.

#### 3.4.6. Cytochrome C

Based on the previously developed concepts of the local mobility of the polypeptide chain, the mechanism of functioning of the Cytochrome C was explained [13,31,32]. The Cytochrome C IS consists of two top-rank ELISes (Figure 21). The cofactor in Cytochrome C is heme. The H18 and M80 residues (structure 1hrc.pdb) interact with the Fe atom of the heme, which stabilizes its position in space. It is important that the M80 residue is part of the site in the Cytochrome C structure with a reduced density of the first rank ELIS—76–83. Low-density ELIS sites of the first rank are labile and potentially mobile. When the elements of the spatial structure corresponding to these ELISes of the highest rank are displaced relative to each other, conformational changes are focused at site 76–83. In this case, the residue M80 is also displaced, which, interacting with the Fe atom, brings it out of the equilibrium state. Apparently, it is the removal of a metal atom from the heme plane that is part of the mechanism of functioning of such heme-containing proteins as hemoglobin or catalase.

It should be noted that both in the case of hydrolytic enzymes and in the case of cytochrome C, the deterministic mobility of the higher rank ELISes is due to the influence of the thermal motion of the surrounding solvent. That is, thermal energy provides the movement of protein fragments corresponding to the highest rank ELIS relative to each other. This movement of the fragments determines the cyclic displacement of the amino acid residues of the active center, ensuring that the protein performs its function. Thus, the proteins in these examples are “anti-entropic molecular machines.”

### 3.5. Identification of Minimal Functional Protein Fragments

#### 3.5.1. Tumour Necrosis Factor (*h*TNF)

In this work [33], the task was to identify the minimum fragment of the *h*TNF protein structure, which is responsible for its function (Figure 22). The *h*TNF protein in active form consists of three identical monomers. Each monomer has a complex spatial organization, the β-structure of which is formed by several β-strands. There were great experimental difficulties in isolating structural fragments. However, during the work, it was identified that the N-terminal part of *h*TNF (residues 3–30) is responsible for its functional activity [33].

#### 3.5.2. Heat Shock Protein (*h*HSP70)

One of the promising methods for therapy of various diseases is the activation of different factors of the innate immune system, including natural killer cells (NK-cells). NK-cells are a special population of lymphocytes from the innate immune system that play an important role in antitumor and antiviral immunity. From the literature it was known that the activation of NK-cells by heat shock protein *h*HSP70 increases the production of γ-interferon (INF-γ). The amino acid sequence of *h*HSP70 consists of 641 residues. The problem was to find a short peptide from *h*HSP70 with an activation effect on NK-cells for further use in clinical practice. 

Preliminary studies have shown that the substrate-binding domain of *h*HSP70 has a desired stimulating effect on the production of γ-interferon by NK-cells. It was found by ANIS method that this fragment corresponds to orange, the highest rank ELIS (G426−M549, Figure 23). However, its length is 123 residues and it was necessary to find a shorter fragment of *h*HSP70 for NK-cells activation.

Interaction site to NK-cells in the substrate-binding domain of *h*HSP70 was unknown. So, the IS of HSP70 sequence was analyzed in detail. It was found that:Fragments 399–408, 411–424, 461–470 and 509–515 are ADD+ sites;Fragment 526–543 is the only one ADD− site in the substrate binding domain sequence of *h*HSP70.

All these fragments were synthesized. In addition the fragment 450–463 (TKD-peptide) was synthesized for functional activity testing. Biological activity of the TKD-peptide is known from the literature. For all synthesized peptides, their effects on production of INF-γ by NK-cells and cytotoxicity were tested by flow cytometry.

One can see that INF-γ production with 526–543 peptide (of ADD− type) twice higher then for 309–408, 411–424, 450–463, 461–470 and 509–515 peptides (ADD+ type) (Figure 23). Thus, the effect of stimulating INF-γ production by NK-cells was obtained only for the peptide of ADD− information type which has the ability to adapt to conformational rearrangements. The results confirmed that the ability of the polypeptide chain to make adaptive conformational rearrangements ensures the formation of effective interactions between polypeptide chains [34,35].

#### 3.5.3. Peptidoglycan Hydrolase gp181 of Bacteriophage φKZ

The Gp181 polypeptide of the φKZ bacteriophage consists of 2237 amino acid residues and contains peptidoglycan hydrolase as a part. The use of the IS analysis method made it possible to identify the hydrolase in the polypeptide sequence [36]. Subsequent testing showed that its catalytic activity was 12 times higher that of egg white lysozyme. These characteristics of the enzyme suggest that it can be used in various biotechnological studies and has high commercial potential.

## 4. Discussion

In the literature, one can find the precise and complete description of protein backbone conformation using libraries of small protein fragments that can approximate every part of protein structures. These libraries, called structural alphabets (SAs), have been widely used in the structure analysis field, from the definition of ligand binding sites to the superimposition of protein structures. SAs are also well suited for analyzing the dynamics of protein structures and their flexibility [37] or finding structural motifs across protein families [38]. Method ANIS instead of SAs does not operate with structural blocks as independent units but identifies areas of correlation in the protein sequence. Some (but not all) rank 1 ELISes, which are pentapeptides and have a stable structure, can be the germs of protein folding and predetermine the folding and structure. Such ELISes may indeed compare to a structural alphabet; however all other regions of the protein, in our opinion, adopt their spatial structure based on the prefolding conformation by the set of folding germs. 

At this stage of our work, we did not set the task of forming any structural alphabets. The paper substantiates the size of an elementary unit of a protein unit based on the entropy criterion. The use of such a new protein unit made it possible to develop a method (ANIS) that allows one to reveal correlations in extended regions of the primary structure of proteins and investigate such areas’ hierarchical structure. Moreover, we believe that constructing structural alphabets should not be based only on the 3D structures of proteins. In our opinion, the elements included in the structural alphabet should preserve their topology in an isolated state. Unfortunately, none of the developers of structural alphabets does topological stability checks. Without such verification, the revealed patterns only reflect the steric capabilities of the polypeptide chain.

Protein folding on the ribosome is distinctly different from the folding of the denatured protein in solution. These differences are determined both by the fact that there is a rate of synthesis of the polypeptide chain and by the folding conditions. During the synthesis on the ribosome, the newly synthesized amino acid residue interacts with the walls of the channel and cavity in the ribosome and with the amino acids of the previously synthesized protein fragment only. In the case of protein folding in solution from the denatured state, there are no interactions with the walls of the ribosome channel and cavity, and all amino acids of the protein are available for interaction.

Further, we will not consider folding on the ribosome but mentally see the process of self-assembly of a protein from a denatured state to a native one in detail. The thought experiment is currently the only possible way to explain the folding process, since experimental methods do not allow considering individual states of protein molecules undergoing the folding process.

At the moment, some patterns of protein folding are known. Now it is believed that this process occurs in stages: at first, the linear protein chain quickly folds up to form a statistical tangle. This is entropic folding [39]. Then a hydrophobic collapse occurs: hydrophobic amino acid residues “hide” deep into the molecule, and hydrophilic ones “settle” along the surface. The result of this stage is the formation of a molten globule. After this, specific bonds are formed, and the protein goes into the state of a true globule, while the free energy drops sharply.

We list below all the facts on which we will build our interpretation of protein folding:

Protein sequences can be considered as overlapping blocks (peptides). A block of five amino acid residues has a minimum entropy function value (Figure 2B) and is called an “information unit” (IU) [6].

Pentapeptides, according to their behavior in molecular dynamics modeling, can be divided into three types: structure-forming, which have a predominant conformational state; trigger, which have two (or three) preferential conformational states; structure-stabilizing, which do not have a predominant conformational state [14].

Hierarchical Elements of Information Structure (ELIS) can be identified by the method of ANalysis of Information Structure (ANIS) in the protein sequence. ELIS are characterized by their position in the sequence of proteins and the hierarchical level (rank) [7].

The interactions between amino acid residues within each individual ELIS are stronger than with amino acid residues outside the considered ELIS. As a rule, ELIS of the highest rank correlate with the position of the structural domains of proteins [29]. So,
(a)ELIS are able to independently fold their spatial structure [33];(b)There is a mobility of spatial elements corresponding to the highest rank ELIS relative to each other in proteins. This mobility provides a molecular mechanism for protein functioning [13,29].

Depending on the density of the first rank ELIS in the protein sequence, the sites can be divided into three different types [10]: 

ADD− sites with abnormal low density of first rank ELIS;

NORMAL sites with normal density of first rank ELIS;

ADD+ sites with abnormal high density of first rank.

ADD+ sites consist of overlapping Structure-Forming peptides (SF-IU), which are very rigid in spatial structure. The ADD− sites mainly consist of structure-stabilizing information units (SS-IU) and are capable of adaptive conformational movements. Thus, the information type of a site reflects its ability to adapt conformational rearrangements, and ADD− sites are the ablest for such rearrangements [10].The fact that in all the studied protein interfaces at least one of the sides contained an ADD− site suggests that a pair of proteins consisting of two ADD+ sites cannot form an “effective” interaction. The ADD− site is a site capable of conformational rearrangements that are necessary for the formation of an “effective” interaction, in which the interacting polypeptide chains are tightly adjacent to each other. This is achieved due to the flexibility of this site, consisting mainly of SS-IUs. We believe that a pair of ADD− and ADD+ sites forms contacts as efficiently as possible since, when forming a contact, only one of the interacting sites is adaptively adjusted. This is exactly how the interfaces in the enzyme-inhibitor and antigen–antibody complexes are arranged. It is in these complexes that the transmission of the signal should be as fast as possible in order to ensure optimal functioning in a living system.

In addition, we note that the only factor providing transitions between different conformational states in the polypeptide chain during protein folding is interaction with solvent molecules due to the Brownian thermal motion [40].

Based on the facts listed above, we offer our own interpretation of the folding process. Our process of folding natural polypeptide chains, similar to the classical description, is a multi-step process as well. The highlighted folding stages are rather arbitrary, but the order of events can be described as follows:The stage of “folding initiation points”. In our opinion, at the very first stage of folding of the polypeptide chain from the denatured state, many turns of the polypeptide chain are formed at the locations of conformationally determined pentapeptides (similar to those found in [14]). These turns become initiation points of the spatial structure spiral elements and turns of various types. In the models proposed earlier, this stage is called the statistical tangle.The stage of “secondary structure germs”. At this stage, the growth of secondary structure elements is observed as an aggregation of structure-forming elements (SF-IU) of ADD+ sites, accompanied by a change in their mutual orientation. There are structure-forming elements that begin, continue or end the elements of the secondary structure. Depending on the combination of such elements in the protein sequence during the formation of helical elements from ADD + sites, the following structural elements are possible:
elements of the secondary structure with “closed” ends, i.e., there are structure-forming pentapeptides passing into a different conformation at the both ends;elements of the secondary structure with “half-open” ends, i.e., there is a structure-forming pentapeptide passing into another conformation at one end of the secondary structure element only;elements of the secondary structure with “open” ends, i.e., there are no structure-forming pentapeptides passing into another conformation at the ends of the secondary structure element.


In addition, ADD+ sites can form non-classical elements of the secondary structure. In every case, ADD+ sites form rigid spatial structures. Thus, ADD+ sites form “germs” of secondary structure elements. At the same stage, a change in the mutual orientation of the “germs” begins.
3.“General topology of the globule” stage. At the previous stages, the “embryos” of the secondary structure elements reached the size of ADD+ sites and their mutual reorientation began. In the process of this reorientation, a state is formed that can be called a “pre-collapse situation”. The “embryos” of the elements of the secondary structure find their position close to the native one. The elements of the secondary structure are being completed using SS-IU from previously unstructured sites. At the same time, conditions are created for the formation of effective interactions with the participation of ADD− sites in the future, i.e., prerequisites are created for the collapse of the spatial structure into a structure with a native topology.4.The “collapse” stage. At this stage, effective contacts are formed with the participation of ADD− sites. In this case, to achieve the minimum energy of the protein globule, changes or even destruction of the elements of the spatial structure or their parts, consisting of ADD− sites, are possible.5.The last stage of folding is the “fitting” stage. At this stage, within the framework of the established topology of the protein globule, the position of the amino acid residues of the side chains is optimized.

Let us mention that some pentapeptides have several (two or three) conformational states—TG-IU. In work [14], we found a small amount of them. In the publication, we did not focus on them, because the data were not enough for analysis. We believe that such peptides can ensure transitions between conformational states in natural protein sequences. Such sites can be called sites with “limited flexibility”. We assume that the primary functional role of such sites is to ensure the catalytic activity of enzymes, although we cannot exclude that they can play the role of switches in the folding processes. It is also important to understand during protein structure analysis, whether a given structural element is “encoded” in the protein sequence or it takes such a conformation only due to the general topology of the protein globule. When designing the protein structure de novo, such regions that do not define the structure during folding can be changed without disrupting the folding process.

This is roughly how the picture of folding looks within the framework of the proposed paradigm. Of course, not all the proteins are capable of renaturation from the denatured state. Here we are discussing only those proteins that are capable of self-assembly. By our concepts, natural polypeptide chains have various flexibility and predetermined conformation along the polypeptide chain. Note that these characteristics determine the local conformation and the ability to form effective interactions between sites remote along the polypeptide chain. It allows nature to “build” the entire “protein molecular machines” set with unique structural and functional characteristics on a single set of elements (amino acid residues). We believe that fragments of the 3D structure of proteins corresponding to ELIS are more stable than other protein fragments of the same length. We have demonstrated by experimental studies on protein design [15,16,17] that removing one ELIS from the primary structure practically does not disrupt the folding of the rest of the protein, i.e., the folding of the ELIS is quasi-independent. The totality of the results allows us to assume that the ANIS method can reveal a hierarchically organized system of “foldons” in protein sequences [41] from 5 to approximately 100 residues (structural domains) long. Probably one of the hierarchy levels of IS corresponds to “foldons” in protein sequences. A series of studies can check and prove these statements. They should show the stability of 3D substructures of the corresponding ELIS at different levels of the hierarchy and experimentally reveal the existence of these 3D substructures in the process of protein folding. 

## 5. Conclusions

The entropy criterion helped to identify the minimal “structural blocks” of natural polypeptide chains. It allowed formulating a new paradigm of the structural organization of natural polypeptide chains, which is based on the idea that the unit of the structural organization of proteins is a group of adjacent amino acid residues.

On the basis of this paradigm, we have developed a method for the ANalysis of Information Structure (ANIS method), which allows us to identify hierarchy in protein sequences—Elements of Information Structure (ELIS). The length of the peptide block at five amino acids is optimal for the ANIS method.

It is shown that the hierarchy in the protein sequence can be revealed in a fairly wide range of the length of the information unit and the number of allowed substitutions. From our point of view, the most optimal length of the information unit is five, and the number of allowed substitutions is one. In addition, a 5-residue fragment of the polypeptide chain may already have stable conformational states of great topological diversity [14], and the natural polypeptide chains themselves are polymers with variable flexibility along the polypeptide chain. 

The density of the first rank ELIS in the protein sequence determines the local ability of the polypeptide chain for adaptive conformational rearrangements of the polypeptide backbone. This property determines the ability of a polypeptide chain to form effective interactions between sites located distantly along the chain or between sites of different polypeptide chains. Thus, the IS of protein sequences provides an important additional criterion for assessing the role of interactions formed between different sites in the polypeptide chain.

It was shown that ELISes of the highest rank are structural domains of proteins or “hidden structural domains”–spatial elements of proteins that play the role of structural domains. It has been shown that the catalytic site residues are distributed over various ELIS of the highest rank [29]. The catalytic center should be located in the area of contact of different ELISes of the highest rank. These data make it possible to carry out the simplest analysis of the mechanisms of functioning of protein molecular machines, which has been demonstrated for a number of metal-containing proteins.

Thus, within the framework of a unified concept, it was possible to solve some fundamentally important problems of protein chemistry. With further development, it should make it possible not only to analyze native proteins, but also to design artificial polypeptide chains with a given spatial organization and, possibly, function. We make such a “daring” statement because the basic principles are already clear, according to which artificial polypeptide chains should be designed, and new theoretical studies and the accumulation of experimental data should allow the development of all the necessary methodological techniques.

## Figures and Tables

**Figure 1 entropy-23-01647-f001:**
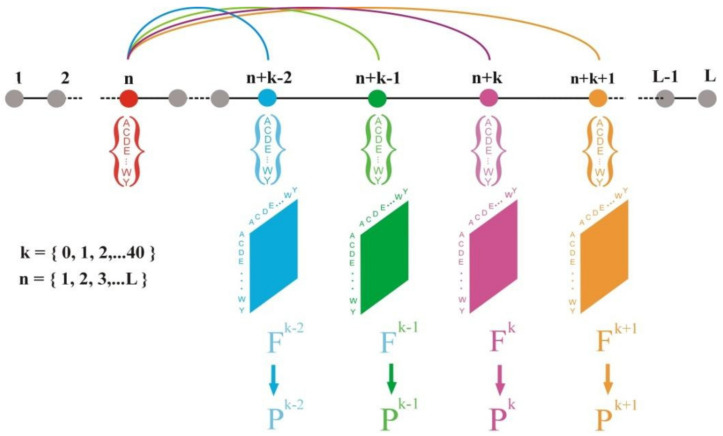
Computation scheme of occurrence frequencies matrices ***F^k^*** and matrices of the probability of occurrence ***P^k^*** of pairs of amino acid residues. ***K*** is a distance between these residues in the protein sequence. The amino acid sequence consists of ***L*** residues. Matrices ***F^k^*** summarize data from all sequences from the protein sequence dataset.

**Figure 2 entropy-23-01647-f002:**
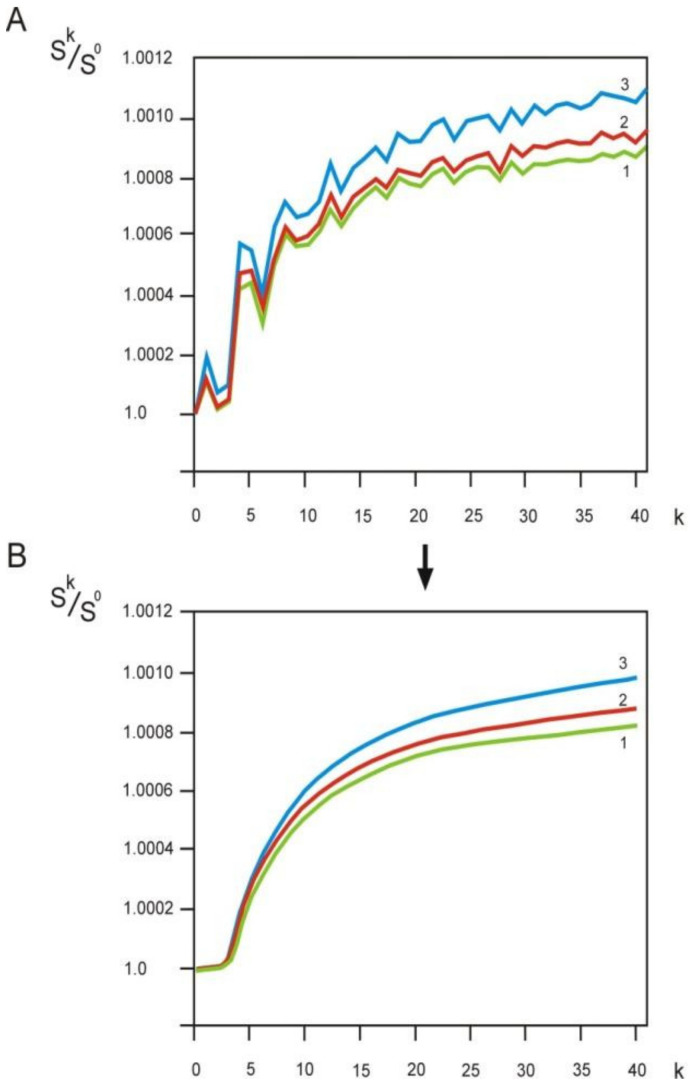
Normalized informational entropy of ***P^k^*** matrices as a function of the distance ***k*** between amino acid residues. (**A**). Dependencies were calculated by the secondary parts of the PIR database. 1. Release 18 of PIR database, 5556 sequences (1,510,026 amino acid residues). 2. Release 27 of PIR database, 12,607 sequences (3,417,043 amino acid residues); 3. Release 49 of PIR database, 58,089 sequences (21,699,210 amino acid residues). (**B**). Normalized informational entropy of matrices ***P^k^*** as a function of the distance ***k*** between amino acid residues without the oscillatory component.

**Figure 3 entropy-23-01647-f003:**
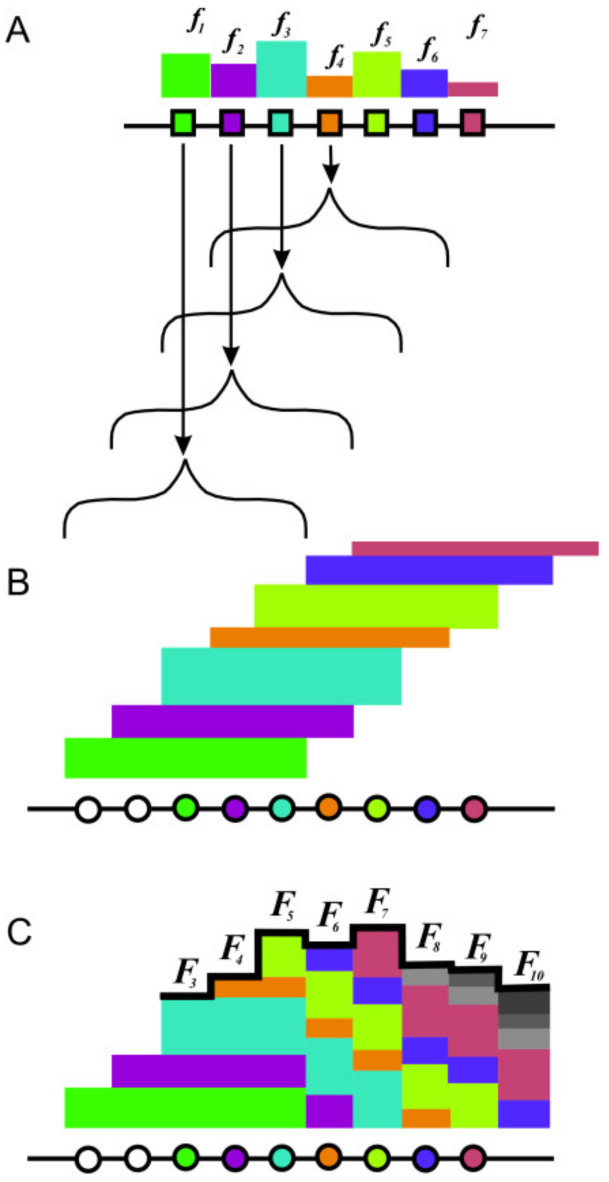
Scheme of the protein population profile ***F_i_*** formation. (**A**). Frequency of occurrence of pentapeptides, calculated as a number of occurrences in a non-redundant large database of protein sequences. (**B**,**C**). Protein population profile summarizes impacts of each pentapeptide.

**Figure 4 entropy-23-01647-f004:**
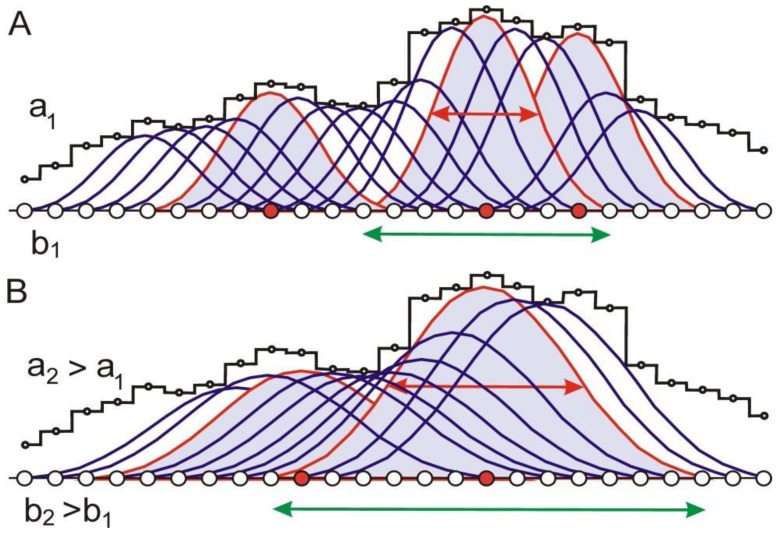
Procedure for nonlinear smoothing of the population profile F (black line). Smoothing functions ***φ(i,a)*** are given for different residues ***i*** for two values of the widths ***a*_1_** (**A**) and ***a_2_*** (**B**) (***a*_2_*> a*_1_*)*** at half height. The smoothing functions ***φ(i,a)***, which are local maxima, are highlighted in gray. The positions of the centers of these functions are marked as red dots on the protein sequence. The red horizontal double-headed arrows show the width ***a*_1_** (**A**) and ***a*_2_** (**B**) of the smoothing function ***φ(i,a)*** at half height. Green horizontal arrows mark the bases ***b*_1_** (**A**) and ***b*_2_** (**B**) of the smoothing functions.

**Figure 5 entropy-23-01647-f005:**
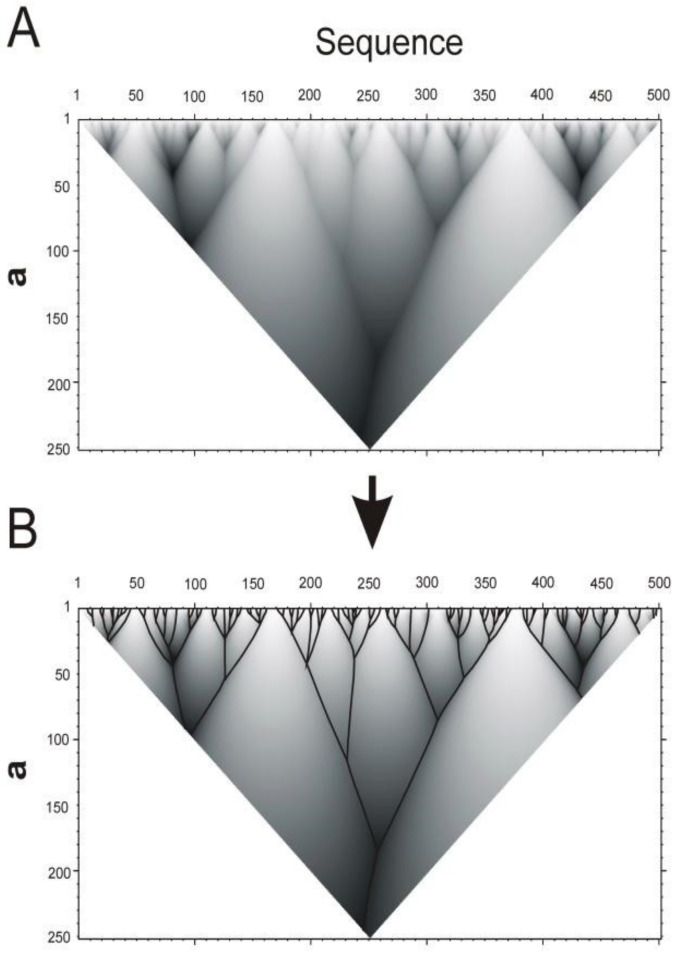
Dependence of the area under the inscribed smoothing function on the amino acid residue ***i*** index and the width ***a*** at half height of the smoothing function ***φ(i,a)*** for catalase from *Micrococcus luteus* sequence P29422 (UniProt code, 500 aa). (**A**). The darker sites correspond to larger areas under the smoothing functions with the same ***a*** values. (**B**). Tree-like graph describing the hierarchical organization of the protein sequence.

**Figure 6 entropy-23-01647-f006:**
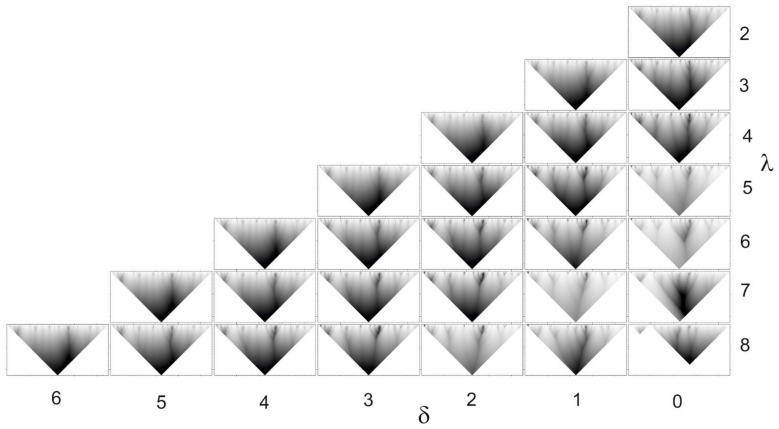
IS of the Q9GZZ6 protein (UniProt) obtained by the ANIS method with different sizes **λ** of “information units” and the number **δ** of allowed substitutions.

**Figure 7 entropy-23-01647-f007:**
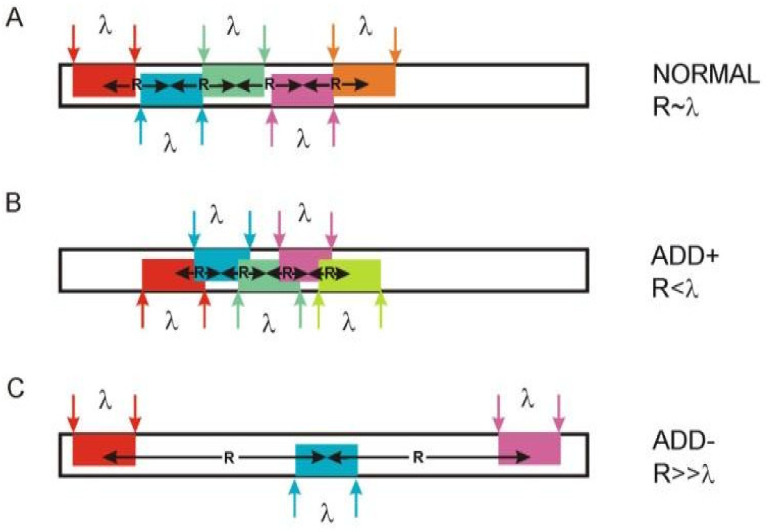
Three types of sites in proteins based on the density of the first rank ELISes in the sequence. Multi-colored rectangles mark the location of the sequence fragments corresponding to the first rank ELIS. The horizontal arrows mark the characteristic distances ***R*** between the central residues of the first rank ELISes. (**A**). NORMAL sites—when the distance between the first rank ELISes is commensurate with the size of the information unit (for pentapeptides, the distance between the centers of first rank ELISes is 5 ± 1 amino acid residues). (**B**). Sites with a high density distribution the first rank ELISes (ADD+ sites). The first rank ELISes overlap in the sequence. The distance between the centers of first rank ELISes for pentapeptides is from 1 to 4 amino acid residues. (**C**). ADD− sites. Distance between the centers of first rank ELISes is significantly greater than the size of the information unit.

**Figure 8 entropy-23-01647-f008:**
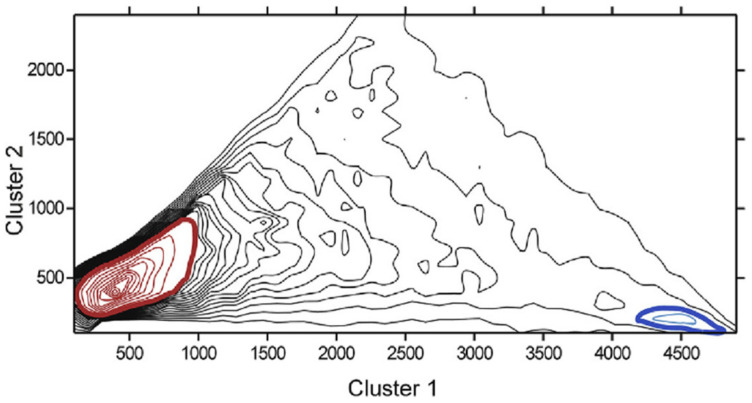
Statistics of 44,860 pentapeptides by the two largest clusters of conformations were obtained in the analysis of the last 5000 conformations of each molecular dynamics trajectories. Of 44,860, 1225 (2.73%) pentapeptides had a stable conformation in which the peptide was at least 80% of the modeling time (dark blue area in the lower right corner). The red area in the lower-left corner corresponds to pentapeptide structures with labile conformation.

**Figure 9 entropy-23-01647-f009:**
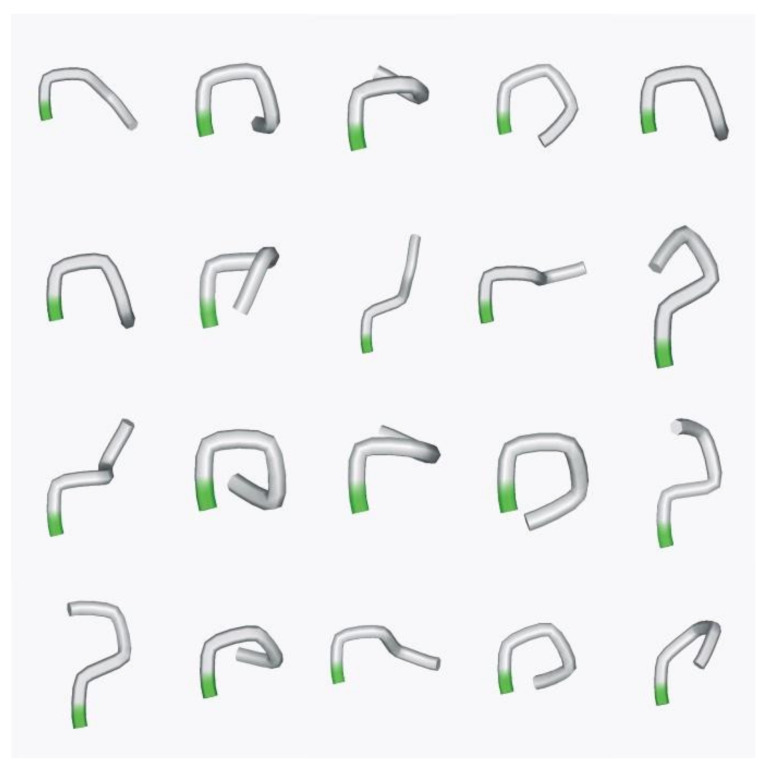
Spatial structures corresponding to the centers of the 20 largest clusters of stable pentapeptides. N-terminal residues in pentapeptides are marked in green.

**Figure 10 entropy-23-01647-f010:**
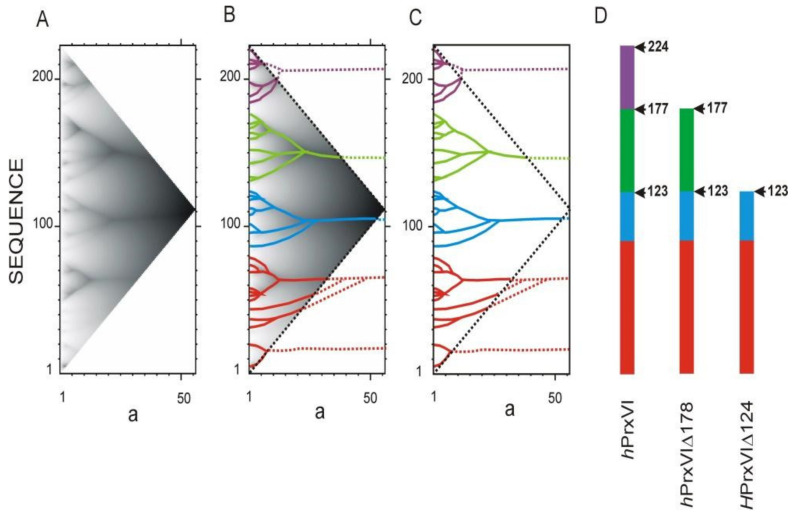
Calculation of the truncated active form of human peroxiredoxin 1-CYS. ***a*** is the width of the smoothing function (see Figure 4). (**A**). IS of human peroxiredoxin 1-CYS *h*PrxVI. (**B**). Allocation of ELISes in the IS. (**C**). Highest rank ELISes in the sequence of human peroxiredoxin 1-CYS. (**D**). Natural and truncated forms of *h*PrxVI [15]. Sequence fragments corresponding to different ELISes of the highest rank are marked with the corresponding color. Arrows mark the boundaries between ELISes of the highest rank.

**Figure 11 entropy-23-01647-f011:**
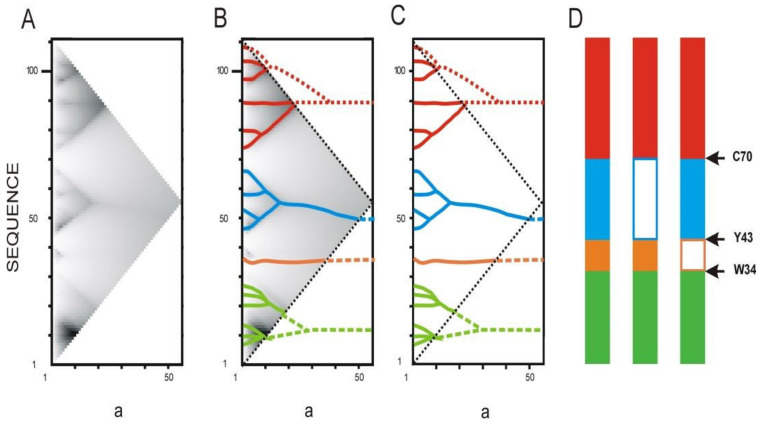
Calculation of the truncated active form of human interleukin-13 *h*IL13. ***a*** is the width of a smoothing function (see Figure 4). (**A**). IS of human interleukin-13 *h*IL13. (**B**). Allocation of ELISes in the IS. (**C**). Highest rank ELISes in the sequence of human interleukin-13 *h*IL13. Sequence fragments corresponding to different ELISes of the highest rank are marked with the corresponding color. (**D**). The hIL13 sequence is represented as ELISes of the highest rank (scheme). Two variants of the recombinant forms of hIL13 were obtained [16]. Arrows mark the boundaries between ELISes of the highest rank.

**Figure 12 entropy-23-01647-f012:**
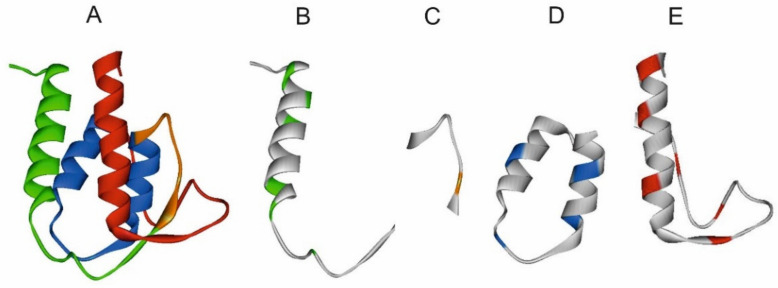
The 3D structure of human interleukin-13 *h*IL13 (1ijz.pdb). (**A**). 3D structure of human interleukin-13 *h*IL13 is colored by ELIS colors. (**B**–**E**). Spatial structures, which correspond to highest rank ELISes in the *h*IL13 sequence (Figure 12D) ((**B**). residues 1–34; (**C**). residues 35–43; (**D**). residues 44–70; (**E**). residues 71–112). Centers of first rank ELISes are shown by appropriate colors.

**Figure 13 entropy-23-01647-f013:**
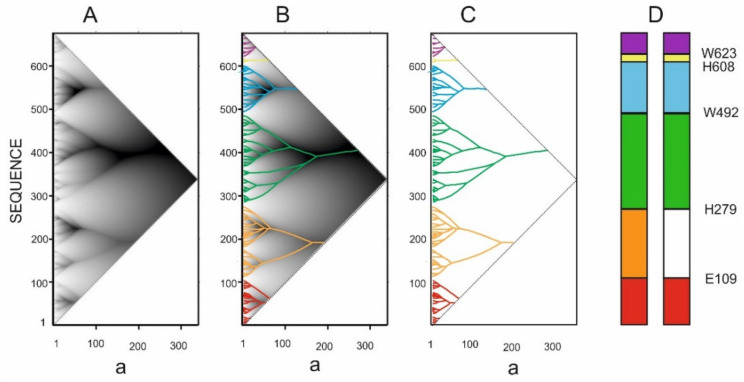
IS of oligopeptidase B from *Serratia proteamaculans* (PSP) sequence: (**A**) function of nonlinear smoothing ***φ(i,a)*** obtained for PSP; ***a*** is a width of the smoothing function (see Figure 4); (**B**) graph of IS of PSP based on function ***φ(i,a)***; (**C**) graph of the PSP IS; (**D**) linear models of PSP sequence with indicated borders between the highest rank ELISes (white region of the higher rank ELIS correspond to Met1-Ser108 fragment).

**Figure 14 entropy-23-01647-f014:**
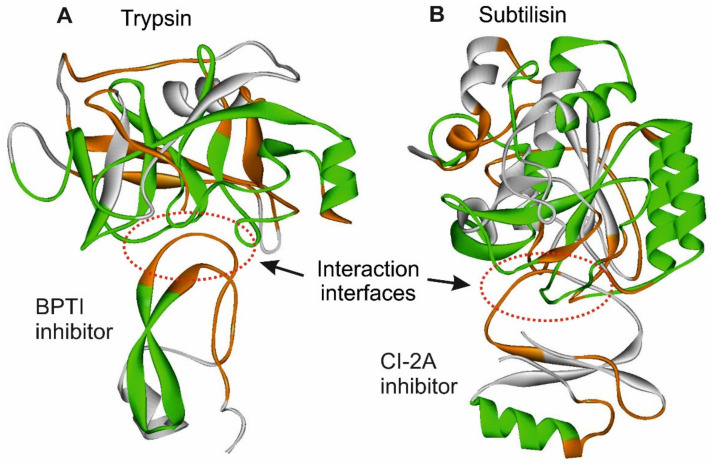
The examples of hydrolytic enzymes complexes with their inhibitors. ADD− sites are shown in green, ADD+ in orange, NORMAL in white. (**A**). Spatial structure of Trypsin/BPTI complex (1CO7.pdb). (**B**). Spatial structure of Subtilisin/CI-2a inhibitor complex (1A10.pdb).

**Figure 15 entropy-23-01647-f015:**
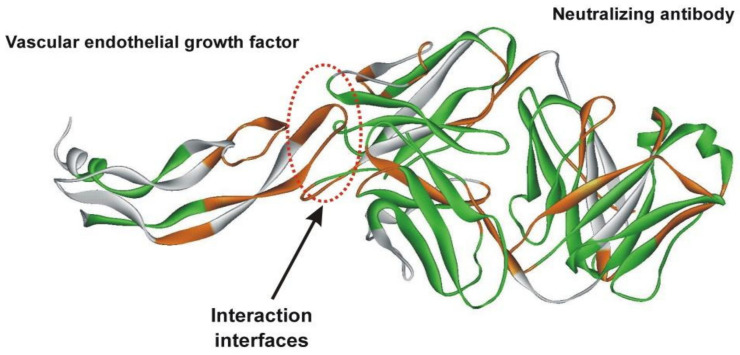
Spatial structure of vascular endothelial growth factor in complex with a neutralizing antibody (1BJ1.PB). ADD− sites are shown in green, ADD+ in orange, NORMAL in white. The protein–protein interface is formed by the ADD+ site of the vascular endothelial growth factor and the ADD− site of the neutralizing antibody.

**Figure 16 entropy-23-01647-f016:**
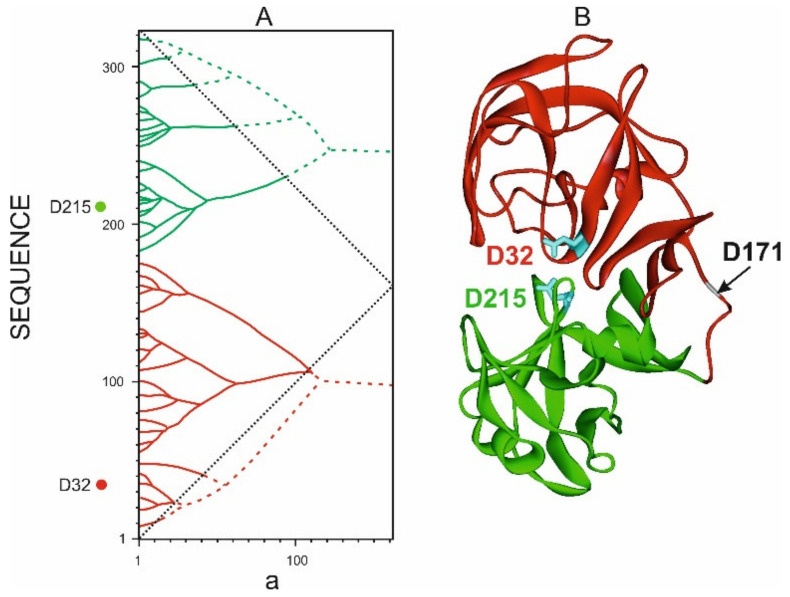
Aspartyl protease pepsin structure (1AM5.PDB). (**A**). Bipartite graph of pepsin IS. (**B**). Two elements of IS corresponding to the highest rank ELIS are highlighted in red and green. Residues ASP32 and ASP215 forming the catalytic center of the enzyme are highlighted in cyan. Amino acid residue D171 divides the pepsin structure into two domains according to X-ray structural analysis [28] (marked with an arrow in the figure).

**Figure 17 entropy-23-01647-f017:**
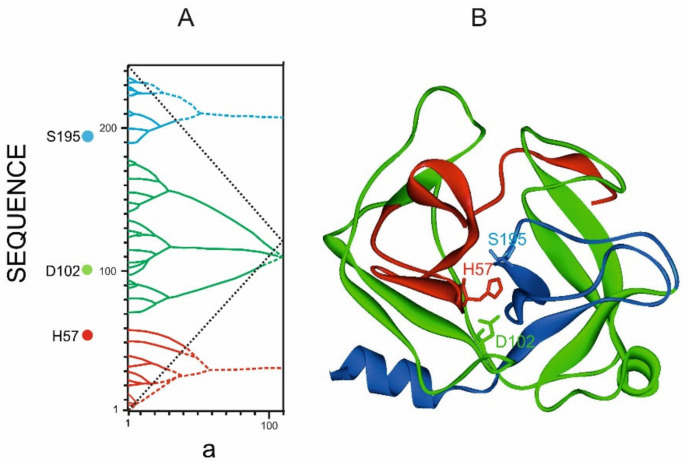
Trypsin structure (1CO7.PDB). (**A**). Tripartite graph of trypsin IS. ***a*** is the width of the smoothing function (**B**). Three fragments of structure corresponding to the highest rank ELIS are highlighted in red, green and blue. In the informational and spatial structures, the positions of the residues of the catalytic center of trypsin H57, D102, and S105 are marked.

**Figure 18 entropy-23-01647-f018:**
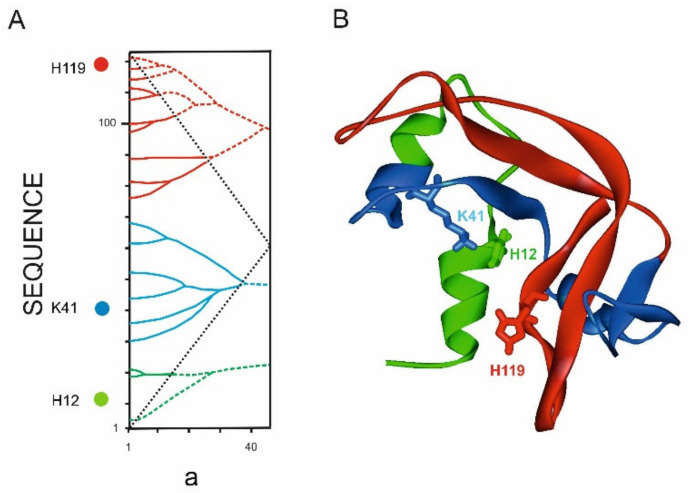
Ribonuclease A structure (1AQP.PDB). (**A**). Tripartite graph of ribonuclease A IS. ***a*** is the width of the smoothing function (**B**). Three fragments of structure corresponding to the highest rank ELIS are highlighted in red, green and blue. In the informational and spatial structures, the positions of the residues of the catalytic center of ribonuclease A H12, K41, and H119 are marked.

**Figure 19 entropy-23-01647-f019:**
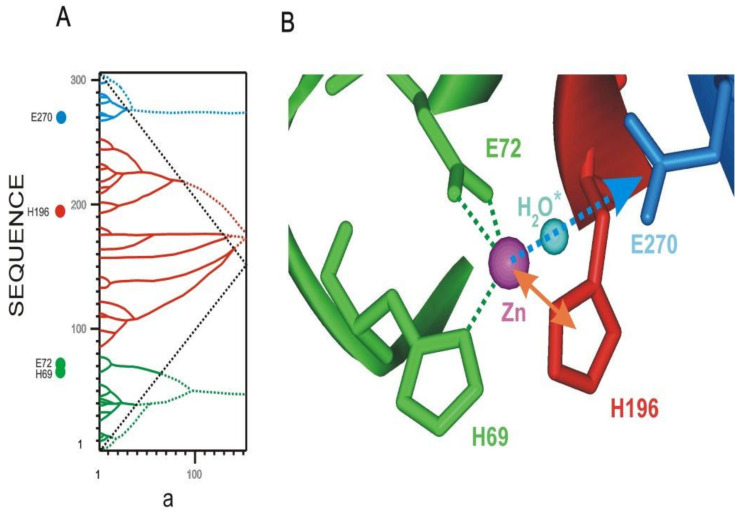
Carboxypeptidase from *Bos taurus* (1m4l.pdb). (**A**). IS of the carboxypeptidase sequence. The highest rank ELISes are highlighted in green, red and blue. ***a*** is the width of the smoothing function (**B**). Structure of the active site of carboxypeptidase. The atom colors correspond to the highest rank ELISes.

**Figure 20 entropy-23-01647-f020:**
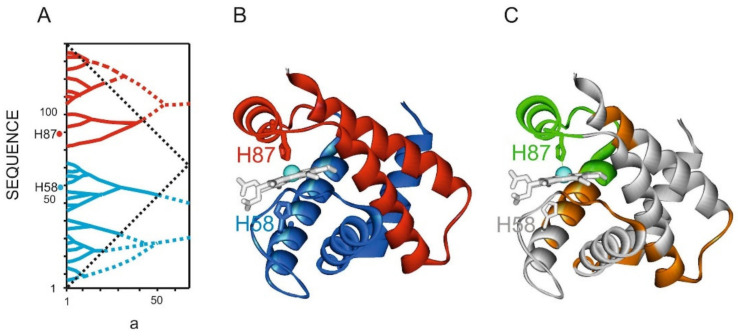
Structure of human hemoglobin alpha-chain (1hz0.pdb). (**A**). IS of hemoglobin alpha-chain sequence. The highest rank ELISes are highlighted in red and blue. ***a*** is the width of the smoothing function (**B**). Structure of the hemoglobin active site. The ribbon colors correspond to the highest rank ELISes. (**C**). H87 residue is part of the ADD− site with a reduced density of the first rank ELIS. The ADD− site (Asn69-Val93) is shown in green, ADD+ sites in orange.

**Figure 21 entropy-23-01647-f021:**
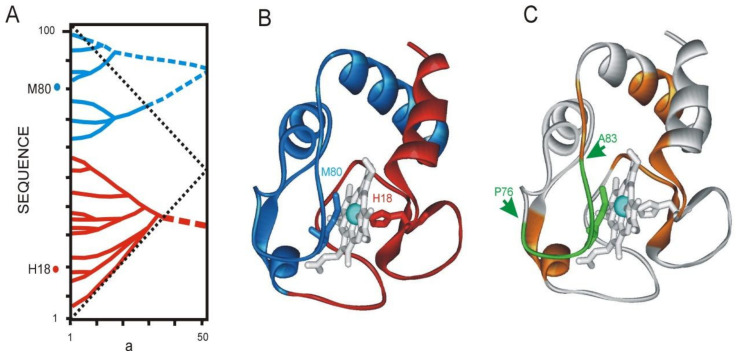
Cytochrome C from *Equus caballus* (1hrc.pdb). (**A**). IS of the cytochrome C sequence. The hierarchically organized highest rank ELISes formed by ANIS method (continuous lines) and by approximations (dashed line). ***a*** is the width of the smoothing function (**B**). His18 and Met 80 residues coordinated to the Fe atom belong to different ELISes of the highest rank (shown in red and blue). (**C**). M80 residue is part of the site with a reduced density of the first rank ELIS—76–83. The ADD− site (P76-A83) with the abnormally low density of first rank ELIS is shown by the arrows. Low-density ELIS sites of the first rank are labile and potentially mobile.

**Figure 22 entropy-23-01647-f022:**
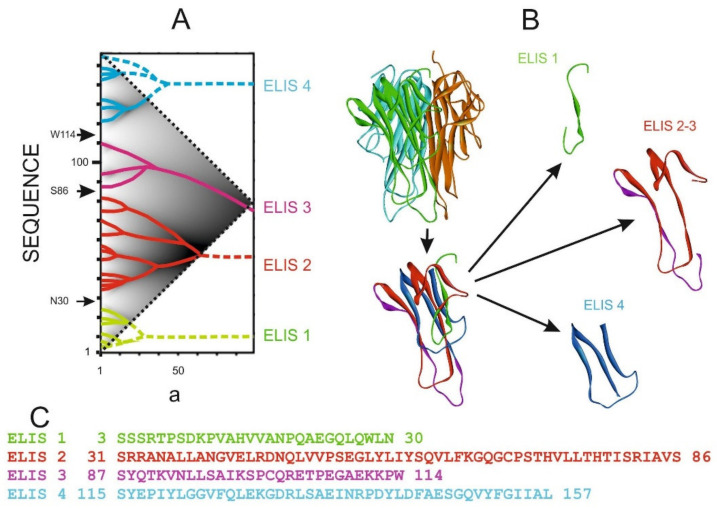
*h*TNF IS. (**A**). The highest rank ELISes are colored in green, orange, violet and blue. The arrows mark the amino acid residues that are the boundaries between the highest rank ELISes. ***a*** is the width of the smoothing function (**B**). Tumor necrosis factor *h*TNF in active form is a homotrimer (1tnf.pdb). Monomer of *h*TNF. Fragments of the *h*TNF structure corresponding to the highest rank ELISes. (**C**). Protein sequence fragments corresponding to the highest rank ELISes.

**Figure 23 entropy-23-01647-f023:**
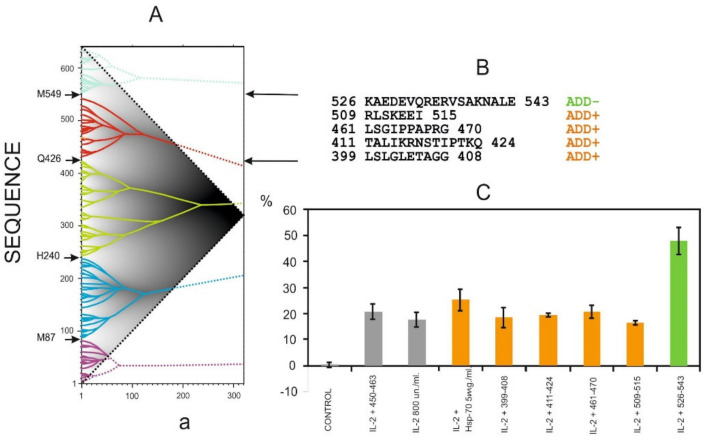
Effect of *h*HSP70 peptide fragments on IFN-γ production by NK cells. (**A**). *h*Hsp70 IS. The highest rank ELISes are shown in different colors. ***a*** is the width of the smoothing function (**B**). Sequences of peptides tested. (**C**). Effect of hHSP70 peptide fragments on IFN-γ production by NK cells, assessed by flow cytometry. NK cells were incubated with IL-2, *h*HSP70 and peptides (2 μg/mL) for 18 h. Unstimulated NK cells were used as a control.

**Table 1 entropy-23-01647-t001:** Possible combinations of the sites in protein–protein interaction interfaces, based on the local density of first rank ELIS in the protein sequence.

Type		Protein 1	Protein 2
I	+−	ADD− ADD+	ADD+ ADD−
II	N−	ADD− NORMAL	NORMAL ADD−
III	−−	ADD−	ADD−
IV	NN	NORMAL	NORMAL
V	N+	NORMAL ADD+	ADD+ NORMAL
VI	++	ADD+	ADD+

**Table 2 entropy-23-01647-t002:** IS characteristics of interaction interfaces in the hydrolase/inhibitor complexes [10]. The percentage of interface amino acids is indicated, %.

Contact Type	TRPS/BPTI		SUBT/CI2A		PME/PMEI		RNAse/RI		PPE/α_1_-RI		MMP3/TIMP		All
**ADD−/ADD+ (%)**	93.6	95.2	39.0	72.7	33.8	85.9	31.4	96.1	14.3	92.9	9.6	62.8	84.2
**ADD−/NORM (%)**	0.0		33.7		31.0		23.5		14.3		46.9		
**ADD−/ADD− (%)**	1.6		0.0		21.1		41.2		64.3		6.3		
**NORM/NORM (%)**	0.0	4.8	11.7	27.3	5.6	14.1	0.0	3.9	0.0	7.1	24.6	37.2	15.8
**ADD+/NORM (%)**	3.2		13.0		8.5		3.9		7.1		12.6		
**ADD+/ADD+ (%)**	1.6		2.6		0.0		0.0		0.0		0.0		

## Data Availability

Not applicable.

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
