# Peer review of "Entropy Analysis of Protein Sequences Reveals a Hierarchical Organization"

_entropy, 2021, doi:10.3390/e23121647_

Round 1

Reviewer 1 Report

Understanding the principles dictating the spatial organizational propensities of amino acids in the protein primary structure has been a hard but interesting problem. The problem in particular stands even more important as there has been growing efforts to come up with synthetic proteins with therapeutic value. Authors in their current work have addressed this problem by using an interesting blend of statistical parameters. They have aimed to demystify the enigma as to how in spite of random distribution of amino acids the properties of the protein are determined by the physicochemical characteristics of amino acids along the polypeptide chain. Authors have inferred that peptide block of five is optimal for studying the structural organization of protein.

 The article is in general a good read with interesting insights but has sections which are little hard to follow owing to 1. Lack of detailing in the technical jargons being used and 2. Too much of dependence on the previous articles which authors have published. This would as a result restrict the readership of the article and would hence not reach out to the broader readership in the field of Protein folding and beyond. I iterate that this is an interesting work but recommend a revision of some sections before acceptance.

Although authors have presented a detailed account of the protein systems which they had earlier studied, a more general contextualization in the discussion is missing. Although the discussion-section presents point by point the series of events which the current approach suggests to constitute the event of protein compaction, a general contextualization with the already known aspects of the folding event is missing. As a result the article would be hard to relate for general protein folding community. I recommend these additions to be made in the discussion section with edits being made in the redundant sections which repeat points already mentioned in the results section.

Authors mention that the method and the results would be valuable for artificial/synthetic proteins but there is hardly any mention of such current efforts or as to how this would be particularly useful to that effort. This is an important piece which must be provided to project the potential of this analysis for a much broader effort.

  1. Abstract section: Background is hard to follow. What is meant by “Analyzing the context of amino acid”? Background should be made more clear and self-explanatory. Although authors in the introduction mentioned “context” relates to the residue-environment, its sudden mention right at the beginning of abstract is little misleading.
  2. Results section of the abstract is very hard to follow for a wider readership. Also the sentences seem to have some disconnect and is not a smooth read. These needs to be addressed.
  3. Although the authors have illustratively explained the philosophy behind “information unit” but they have not connected with the existing concepts like “foldon”. They should discuss the “information-unit” from the perspectives of “foldon” in the discussion.
  4. Line 68: “These blocks are similar for some residues.”: Earlier authors showed/mentioned the residues were constituent of three blocks. This sentence is little hard to follow. Please re-phrase.
  5. Line 84: “we increase the depth of the 84 evolutionary consideration of the pentapeptide”. This region is hard to comprehend. What is meant by "increase the depth of the evolutionary consideration"?
  6. Line 719: “Further, we will not consider folding on the ribosome, but mentally consider in detail the process of self-assembly of a protein from a denatured state to a native one.” -- This sentence is very strangely written. Please re-write.
  7. Line 733: “A block of five amino acid residues has a minimum of entropy and was called an "information unit" (IU)” : Is minimum entropy translated as "information unit"?
  8. Line 768-769: “We believe that a pair of ADD- and ADD + sites forms contacts as efficiently as possible since when forming a contact, only one of the interacting sites is adaptively adjusted.”: This would be interesting to analyse in the context of some probable co-evolutionary signal.
  9. Line 772: What do the author mean by “control signal”?
  10. Line 785: Did you observe any experimental correlation with the "statiscal tangle" and reported "foldons"?
  11. Line 787: “Secondary structure Embryo” -- How the "secondary structure embryo" can have a plausible explanation for conformational compaction events observed in protein with signification extents of disordered segments?
  12. Line 826 to 829: The authors could refer to proteins with reported active site information. What is meant by "switches"? Conformational switching leading to some transient structures during the folding process?

Author Response

Understanding the principles dictating the spatial organizational propensities of amino acids in the protein primary structure has been a hard but interesting problem. The problem in particular stands even more important as there has been growing efforts to come up with synthetic proteins with therapeutic value. Authors in their current work have addressed this problem by using an interesting blend of statistical parameters. They have aimed to demystify the enigma as to how in spite of random distribution of amino acids the properties of the protein are determined by the physicochemical characteristics of amino acids along the polypeptide chain. Authors have inferred that peptide block of five is optimal for studying the structural organization of protein.

 The article is in general a good read with interesting insights but has sections which are little hard to follow owing to 1. Lack of detailing in the technical jargons being used and 2. Too much of dependence on the previous articles which authors have published. This would as a result restrict the readership of the article and would hence not reach out to the broader readership in the field of Protein folding and beyond. I iterate that this is an interesting work but recommend a revision of some sections before acceptance.

Although authors have presented a detailed account of the protein systems which they had earlier studied, a more general contextualization in the discussion is missing. Although the discussion-section presents point by point the series of events which the current approach suggests to constitute the event of protein compaction, a general contextualization with the already known aspects of the folding event is missing. As a result the article would be hard to relate for general protein folding community. I recommend these additions to be made in the discussion section with edits being made in the redundant sections which repeat points already mentioned in the results section.

Authors mention that the method and the results would be valuable for artificial/synthetic proteins but there is hardly any mention of such current efforts or as to how this would be particularly useful to that effort. This is an important piece which must be provided to project the potential of this analysis for a much broader effort.

The authors thank the reviewer for carefully reading the manuscript and appreciating the work.

  1. Abstract section: Background is hard to follow. What is meant by “Analyzing the context of amino acid”? Background should be made more clear and self-explanatory. Although authors in the introduction mentioned “context” relates to the residue-environment, its sudden mention right at the beginning of abstract is little misleading.

Corrected. We have removed this term in the beginning of the abstract.

  1. Results section of the abstract is very hard to follow for a wider readership. Also the sentences seem to have some disconnect and is not a smooth read. These needs to be addressed.

Done. We have updated the Results section of the abstract.

  1. Although the authors have illustratively explained the philosophy behind “information unit” but they have not connected with the existing concepts like “foldon”. They should discuss the “information-unit” from the perspectives of “foldon” in the discussion.

Thank you very much, we have been added paragraph with “foldon” concept speculations in the Discussion.

  1. Line 68: “These blocks are similar for some residues.”: Earlier authors showed/mentioned the residues were constituent of three blocks. This sentence is little hard to follow. Please re-phrase.

Done

  1. Line 84: “we increase the depth of the 84 evolutionary consideration of the pentapeptide”. This region is hard to comprehend. What is meant by "increase the depth of the evolutionary consideration"?

This sentence has been rephrased.

  1. Line 719: “Further, we will not consider folding on the ribosome, but mentally consider in detail the process of self-assembly of a protein from a denatured state to a native one.” -- This sentence is very strangely written. Please re-write.

Done

  1. Line 733: “A block of five amino acid residues has a minimum of entropy and was called an "information unit" (IU)” : Is minimum entropy translated as "information unit"?

No, minimum entropy does not translate as "information unit". This sentence has been rephrased.

  1. Line 768-769: “We believe that a pair of ADD- and ADD + sites forms contacts as efficiently as possible since when forming a contact, only one of the interacting sites is adaptively adjusted.”: This would be interesting to analyse in the context of some probable co-evolutionary signal.

Yes, we are going to learn this question in the future.

  1. Line 772: What do the author mean by “control signal”?

We rephrased this sentence. We meant that it is a signal that starts some process or cascade.

  1. Line 785: Did you observe any experimental correlation with the "statiscal tangle" and reported "foldons"?

Experimental studies on identifying foldons in the process of self-organization of polypeptide chains were not carried out in this work. However, the presented data of experiments on protein design allow us to assume that the hierarchical structures revealed by the ANIS method probably correspond to the hierarchical system of foldons. An indirect confirmation of this statement is that the removed protein fragments do not disrupt the folding of the rest of the natural polypeptide chains.

  1. Line 787: “Secondary structure Embryo” -- How the "secondary structure embryo" can have a plausible explanation for conformational compaction events observed in protein with signification extents of disordered segments?

You're right, "secondary structure embryo" can not give a plausible explanation for disordered protein folding. We pointed out at the beginning of the Conclusion that this reasoning is applicable only for proteins capable of self-assembly.

  1. Line 826 to 829: The authors could refer to proteins with reported active site information. What is meant by "switches"? Conformational switching leading to some transient structures during the folding process?

We have been added this explanation into the text:

We found a few pentapeptides in work [14] with several (two or three) conformational states. In the publication, we did not focus on them because the data were not enough for analysis. We believe that in natural protein sequences, such pentapeptides can make transitions between conformational states. Such sites can be called sites with "limited mobility". We assume that the primary functional role of such sites is to ensure the catalytic activity of enzymes, although we cannot exclude that they can play the role of switches in the folding processes.

Reviewer 2 Report

In the manuscript titled “Entropy Analysis of Protein Sequences Reveals a Hierarchical Organization”, authors performed statistical analysis on the amino acid sequence space of proteins and revealed abstract trends in the way several amino acids appear in the sequence space, their hierarchy in time and how these contain further information regarding interactions in a protein. In addition, they showed how this information can be used as a starting point for the design of polypeptide chains with target interactions. The study described in the manuscript is interesting and should be relevant to the community.

However, I have some concerns about the use of several assumptions key to the results. I will outline my concerns in detail and some other suggestions regarding the manuscript below:

  1. The science behind the study can be immensely benefited if authors describe the physical interpretation behind the informational entropy. For example, it is not very clear what to interpret from Fig. 2. Authors show that block size goes up, the relative entropy also increases. However, then they fail to comment on why it does so. Also, the interpretation of the oscillatory components in the Fig. 2A is underwhelming. It might be further useful to show how Fourier analysis was performed to obtain the oscillatory frequencies and how do these frequencies correspond to the secondary structure.

  2. In Fig2B, it should be useful to show how the oscillatory components from the curves are removed?

  3. In lines 151-155, authors mention that higher the number of protein sequences the higher is the value of normalized information entropy. It is not clear how that relationship is obtained given that in Fig. 2B, the blue curve (3) has highest entropy but has the lowest number of sequences (5556). In addition, why is this relationship expected.

  4. In Fig 4, the highlighted gray region is not visible (or it seems like a shaded blue region).

  5. In Fig 6, why does the plot corresponding to δ- 0 and λ-8 does not start from the bottom of y axis (i.e. rank 1)

  6. It seems that in Fig 7, the authors have described the three types of sites in a protein based on their size of information unit and the distance between them. In Panels A and B, the meaning of R is not shown on the schematic diagrams. Also, in panel C, the description on the right does not look right. In addition, in lines 281-284, the assumptions behind the mobility and the size of the information unit is not clear. These might need to be more justified.

  7. The simulation protocol described in lines 299-303 is insufficient. Please add further details of force-fields, initial conditions, types of thermostat, etc used. In addition, 10ps simulation for each sequence seems awfully small. A further analysis of convergence should be added to check if results are reproducible.

  8. In lines 332-348, the authors attempt to connect the rank of ELISes with the structural domains. The arguments shown need to be substantiated with figures and proofs.

  9. In addition to these points, the manuscript should be benefited with another grammar and punctuation check.

Author Response

In the manuscript titled “Entropy Analysis of Protein Sequences Reveals a Hierarchical Organization”, authors performed statistical analysis on the amino acid sequence space of proteins and revealed abstract trends in the way several amino acids appear in the sequence space, their hierarchy in time and how these contain further information regarding interactions in a protein. In addition, they showed how this information can be used as a starting point for the design of polypeptide chains with target interactions. The study described in the manuscript is interesting and should be relevant to the community.

However, I have some concerns about the use of several assumptions key to the results. I will outline my concerns in detail and some other suggestions regarding the manuscript below:

  1. The science behind the study can be immensely benefited if authors describe the physical interpretation behind the informational entropy. For example, it is not very clear what to interpret from Fig. 2. Authors show that block size goes up, the relative entropy also increases. However, then they fail to comment on why it does so. Also, the interpretation of the oscillatory components in the Fig. 2A is underwhelming. It might be further useful to show how Fourier analysis was performed to obtain the oscillatory frequencies and how do these frequencies correspond to the secondary structure.

The main conclusion from the dependence shown in Figure 2B is the lowerest and constant value of informational entropy for short distances between residues. It means that residues are strongly correlated. This made it possible to determine the optimal size of the "information unit". We carried out Fourier analysis and removal of the vibrational component in the R package. We added a mention of this to the text.

  1. In Fig2B, it should be useful to show how the oscillatory components from the curves are removed?

This is a standard function in the R and this technical detail is hardly of interest to readers.

  1. In lines 151-155, authors mention that higher the number of protein sequences the higher is the value of normalized information entropy. It is not clear how that relationship is obtained given that in Fig. 2B, the blue curve (3) has highest entropy but has the lowest number of sequences (5556). In addition, why is this relationship expected.

Thanks for your observation, the figure captions were confused. Fixed.

  1. In Fig 4, the highlighted gray region is not visible (or it seems like a shaded blue region).

You are quite right, it looks like a shaded blue area, but it is actually gray.

  1. In Fig 6, why does the plot corresponding to δ- 0 and λ-8 does not start from the bottom of y axis (i.e. rank 1)

The plot corresponding to δ- 0 and λ-8 starts from rank 1, at the top. With the size of the information unit λ = 8 amino acid residues in the Q9GZZ6 sequence, not all octapeptides were found in the NRDB90 protein sequence database at least once. As a result, when calculating the information structure, a gap arises, which leads to a decrease in the area of definition of the smoothing function φ (i, a) (Figure 6, λ = 8, δ = 0).

  1. It seems that in Fig 7, the authors have described the three types of sites in a protein based on their size of information unit and the distance between them. In Panels A and B, the meaning of R is not shown on the schematic diagrams. Also, in panel C, the description on the right does not look right. In addition, in lines 281-284, the assumptions behind the mobility and the size of the information unit is not clear. These might need to be more justified.

You're right, the caption on the right in Figure 7 in panel C is incorrect. Fixed. In lines 281-284 the assumptions behind the mobility and the size of the information unit are re-phrased.

  1. The simulation protocol described in lines 299-303 is insufficient. Please add further details of force-fields, initial conditions, types of thermostat, etc used. In addition, 10ps simulation for each sequence seems awfully small. A further analysis of convergence should be added to check if results are reproducible.

We published all details about molecular dynamics procedure in [14]. There is a reference in the text.

  1. In lines 332-348, the authors attempt to connect the rank of ELISes with the structural domains. The arguments shown need to be substantiated with figures and proofs.

The relationship between the highest rank ELIS and structural domains has been established only on several examples [7,10]. At the moment, we have received information about the structural organization of the ELIS at various levels of the hierarchy. This data will be analyzed and presented in our following publications.

  1. In addition to these points, the manuscript should be benefited with another grammar and punctuation check.

Thank you, done.

Reviewer 3 Report

The authors present an article that analyze the environment of amino acids in protein sequences. This work continues the cycle of author’s works devoted to the analysis of protein.

Using informational entropy of protein sequences, the authors present a method ANIS (Analysis of Informational Structure) that find block of five adjacent amino acid residues called “information unit”.

Authors proposed a number of practical applications based of their method.

The manuscript is properly written. But I have some remarks or questions:

1/ The authors used the word “Structure” in different definitions (ANIS, ELIS, etc…) but their study is not related to the protein structure in term of 2D/3D structures. The authors should clarify this point in the manuscript to avoid confusion.

2/ In materials and methods, line 100.

The part of the sentence “significant level of homology” is biologically wrong. Sequences are homologous or not, we cannot have level of homology (level of similarity, yes !).

3/ The figure 6 is not easy to read and understand.

4/ dynamics study of model pentapeptides

A lot of details are missing considering MD simulations (Force Field used, MD parameters, …)

5/ line 426 : I did not understand very well the sentence “We assume that all first rank ELISes are turns, but not all turns are first rank ELISes.”

6/ I did not find in the discussion the comparison of their method with other methods splitting the sequence in structural blocks as the structural alphabets (for example doi: 10.3389/fmolb.2015.00020 or doi: 10.1093/nar/gkq478) The authors should discuss more this point in the manuscript.

Author Response

The authors present an article that analyze the environment of amino acids in protein sequences. This work continues the cycle of author’s works devoted to the analysis of protein.

Using informational entropy of protein sequences, the authors present a method ANIS (Analysis of Informational Structure) that find block of five adjacent amino acid residues called “information unit”.

Authors proposed a number of practical applications based of their method.

The manuscript is properly written. But I have some remarks or questions:

  1. The authors used the word “Structure” in different definitions (ANIS, ELIS, etc…) but their study is not related to the protein structure in term of 2D/3D structures. The authors should clarify this point in the manuscript to avoid confusion.

Thank you, we reduced number of “structures” in the text, done.

  1. In materials and methods, line 100.

The part of the sentence “significant level of homology” is biologically wrong. Sequences are homologous or not, we cannot have level of homology (level of similarity, yes !).

Thank you, fixed.

  1. The figure 6 is not easy to read and understand.

We did not find a way to improve figure 6 for better reading and understanding.

  1. dynamics study of model pentapeptides A lot of details are missing considering MD simulations (Force Field used, MD parameters, …)

We published all details about molecular dynamics procedure in [14]. There is a reference in the text.

  1. line 426 : I did not understand very well the sentence “We assume that all first rank ELISes are turns, but not all turns are first rank ELISes.”

We corrected the text for better understanding.

  1. I did not find in the discussion the comparison of their method with other methods splitting the sequence in structural blocks as the structural alphabets (for example doi: 10.3389/fmolb.2015.00020 or doi: 10.1093/nar/gkq478). The authors should discuss more this point in the manuscript.

Thank you for this valuable comment! We added the text below to the Discussion:

In the literature, one can find the precise and complete description of protein backbone conformation using libraries of small protein fragments that can approximate every part of protein structures. These libraries, called structural alphabets (SAs), have been widely used in the structure analysis field, from the definition of ligand binding sites to the superimposition of protein structures. SAs are also well suited for analyzing the dynamics of protein structures and their flexibility [40] or finding structural motifs across protein families [41]. Method ANIS instead of SAs does not operate with structural blocks as independent units but identifies areas of correlation in the protein sequence. Some (but not all) rank 1 ELISes, which are pentapeptides and have a stable structure, can be the germs of protein folding and predetermine these folding and structure. Such ELISes may indeed compare to a structural alphabet. But all other regions of the protein, in our opinion, adopt their spatial structure based on the prefolding conformation by the set of folding germs.

At this stage of our work, we did not set the task of forming any structural alpha-bets. The paper substantiates the size of an elementary unit of a protein unit based on the entropy criterion. The use of such a new protein unit made it possible to develop a method (ANIS) that allows one to reveal correlations in extended regions of the primary structure of proteins and investigate such areas' hierarchical structure. Moreover, we believe that constructing structural alphabets should not be based only on the 3D structures of proteins. In our opinion, the elements included in the structural alphabet should preserve their topology in an isolated state. Unfortunately, none of the developers of structural alphabets does topological stability checks. Without such verification, the revealed patterns only reflect the steric capabilities of the polypeptide chain.

Round 2

Reviewer 2 Report

The changes made by the author are appropriate and sufficient. The manuscript should ready for publication.